# DAViD: Domain Adaptive Visually-Rich Document Understanding with Synthetic Insights

## Abstract

Visually-Rich Documents (VRDs), encompassing elements like charts, tables, and references, convey complex information across various fields. However, extracting information from these rich documents is labor-intensive, especially given their inconsistent formats and domain-specific requirements. While pretrained models for VRD Understanding have progressed, their reliance on large, annotated datasets limits scalability. This paper introduces the Domain Adaptive Visually-rich Document Understanding (DAViD) framework, which utilises machine-generated synthetic data for domain adaptation. DAViD integrates fine-grained and coarse-grained document representation learning and employs synthetic annotations to reduce the need for costly manual labelling. By leveraging pretrained models and synthetic data, DAViD achieves competitive performance with minimal annotated datasets. Extensive experiments validate DAViD's effectiveness, demonstrating its ability to efficiently adapt to domain-specific VRDU tasks.

## 1 Introduction

In today's information-driven world, documents with complex visual structures, such as charts, tables, and references, are vital tools for conveying detailed ideas. These Visually-Rich Documents(VRDs) are commonly used across various domains, offering crucial insights backed by expertise. However, manually extracting relevant information from the vast number of VRDs available is an overwhelming and inefficient process, particularly in fields where domain-specific knowledge is critical. The task becomes even more complex due to the variability in document formats, especially given the rapidly increasing demands across multiple domains such as finance(Ding et al., 2023), education(Wang et al., 2021), and politics(Wang et al., 2023), academic papers(Ding et al., 2024b). VRDs often exhibit flexible and inconsistent layouts, making extracting accurate information a significant challenge. From a human perspective, understanding a document in a new domain begins by examining its format and layout, followed by a detailed analysis of its content in response to user demands. Several pretrained large frameworks for VRD Understanding (VRDU), such as LayoutLMv3(Huang et al., 2022) and StructExtv3(Lyu et al., 2024), have emerged, leveraging self-supervised learning to capture general document structures. While these models show promise, their practical application in specialized domains still relies heavily on large, annotated datasets tailored to the domain in question. Creating high-quality annotations demands expert knowledge and extensive effort, particularly when deciphering these documents' logical arrangement and structure. While PDF parsers and OCR tools can extract initial structural data—such as text lines or boxes—high-quality layout annotations often require additional expert-guided processing, using source files like XML or HTML to refine the extracted data. This bottleneck delays the deployment of VRDU models and limits their practical scalability across diverse fields.

Beyond document structure, understanding document content also presents significant challenges. Task-oriented datasets with detailed annotations are typically needed to train models for effective information extraction or question-answering tasks, particularly in domains requiring specific expertise, such as finance, academia, or receipts. Annotating these documents requires an expert understanding of their content and frequently involves preliminary layout annotations. This reliance on expert annotations can hinder the deployment of VRDU models in real-world scenarios due to the labor-intensive nature of the process. Recent advances in large language models (LLMs) and multimodal large models (MLMs) have demonstrated promising zero-shot performance on VRDU tasks by leveraging extensive training on varied corpora. These models can even be prompted to

generate synthetic VRD-QA datasets, potentially reducing the need for manual annotations. However, translating this capability into practical, real-world applications remains challenging. In response to these challenges, this paper proposes a novel approach that leverages machine-generated synthetic data to enable domain adaptation for Visually-Rich document understanding. By utilizing synthetic data to bridge the gap between general and domain-specific documents from VRD structure and content perspectives, we aim to significantly reduce the need for costly expert annotations. This approach offers a promising solution for applying VRDU models in a more scalable and efficient manner across various domains without compromising the accuracy of information extraction.

This paper introduces the Domain Adaptive Visually-Rich Document Understanding (DAViD) framework, a novel VRDU approach that utilizes machine-generated synthetic data for domain understanding enhancement. DAViD is designed to achieve high performance in document understanding tasks, even with limited annotated documents, by leveraging pretrained models from general domains and introducing effective domain adaptation strategies. The framework incorporates fine-grained (token-level) and coarse-grained (document entity-level) processing to enrich document representations while addressing domain-specific challenges through machine-generative synthetic data. By automatically generating synthetic annotations, DAViD reduces the dependence on expert-labeled datasets while maintaining high extraction accuracy for VRDU.

The key contributions of this paper are as follows: **1) Joint-grained VRDU Framework**: We present DAViD, a framework that integrates fine-grained (token-level) and coarse-grained (document entity-level) document representations, leveraging pretrained models and synthetic data to achieve competitive performance with minimal annotations. **2) Synthetic Data Generation Workflow**: We propose a workflow that generates structural and semantic annotations using off-the-shelf tools and LLMs, significantly reducing manual annotation efforts and making the VRDU process scalable. **3) Domain Adaptation Strategies**: We introduce strategies within DAViD to bridge the gap between general and domain-specific documents, enabling robust performance across new domains without extensive domain-specific training data. **4) Comprehensive Validation**: Extensive experiments demonstrate that DAViD performs comparably to models trained on large annotated datasets, effectively adapting to domain-specific VRDU tasks using synthetic data.

## 2 RELATED WORK

### 2.1 VISUALLY-RICH DOCUMENT UNDERSTANDING

Heuristic methods(Watanabe et al., 1995; Seki et al., 2007; Rusinol et al., 2013) and statistical machine learning (Oliveira & Viana, 2017) were applied to closed-domain document applications, but required expert customization. Recent advances in deep learning, including models based on LSTM and CNN(Katti et al., 2018; Denk & Reisswig, 2019; Zhao et al., 2019), feature-driven approaches(Yu et al., 2021; Zhang et al., 2020; Wang et al., 2021), and layout-aware pre-trained frameworks(Xu et al., 2020; Wang et al., 2022; Hong et al., 2022), have shown promise in enhancing document representation, but rely heavily on extensive, well-annotated data for domain-specific knowledge transfer. Visual-cues integrated pretrained frameworks(Xu et al., 2021; Huang et al., 2022) aim to generate more comprehensive document representations but are limited in capturing long-term logical relationships. Recently, joint-grained frameworks(Yu et al., 2022; Lyu et al., 2024) have emerged to address these challenges but face issues with heavy fine-tuning, similar to other deep learning frameworks. Large Language Model (LLM)-based frameworks(He et al., 2023; Fujitake, 2024; Luo et al., 2024) have improved zero-shot performance for document understanding tasks by leveraging broad pretraining. However, they still require extensive training and data to perform effectively in specific domains. The reliance on large-scale, annotated datasets remains a barrier, underscoring the need for scalable solutions like synthetic data generation, as explored in this paper.

### 2.2 DOMAIN ADAPTATION AND KNOWLEDGE DISTILLATION

Domain adaptation is crucial in transfer learning, encompassing several variants such as unsupervised domain adaptation(Wang et al., 2020) and source-free domain adaptation(Liang et al., 2020), which focus on transferring knowledge from one source domain to a target domain that differs from our scenarios. Another subproblem within transfer learning, knowledge distillation(Hinton et al., 2015), involves transferring knowledge from a large-scale teacher to a small student networks. This has

been widely applied in language (Adhikari et al., 2020), vision (Fang et al., 2021), and multimodal applications (Ma et al., 2023), yet there is a lack of research exploring knowledge distillation in Visually-Rich Document Understanding (VRDU). While some efforts, such as (Ding et al., 2024c), have explored joint-grained knowledge distillation for VRDU, they rely heavily on large, annotated datasets and require extensive fine-tuning for practical use. Our work addresses this gap by proposing a novel approach that utilizes synthetic data to enable domain adaptation and distillation, achieving competitive results without needing large-scale manual annotations.

## 3 PROBLEM FORMULATION

**Preliminary Definition** Given a collection of documents $\mathbb{D} = \{D_1, D_2, \ldots, D_m\}$ from a specific domain containing $m$ documents, the purpose of the task is to extract the predefined $k$ types of key information $\mathbb{Y} = \{Y_1, Y_2, \cdots, Y_k\}$ from $\mathbb{D}$. The entire document collection can be divided into three subsets $\mathbb{D} = \{\mathbb{D}_n, \mathbb{D}_g, \mathbb{D}_i\}$, including a relatively larger unannotated set $\mathbb{D}_n$, a relatively small manually annotated guidance set $\mathbb{D}_g$, and $\mathbb{D}_i$ a set containing practical inference cases of arbitrary size. Following the setting up of the joint-grained frameworks, (Gu et al., 2021; Ding et al., 2024c), a document $D \in \mathbb{D}$ has fine/coarse-grained information. Fine-grained information from a document $D$ is represented by a sequence of textual tokens, where $T_D = \{t_1, t_2, \cdots, t_n\}$ with text content and the coordinates of the box of the bounding of each token, $t = (text, box)$. $D$ also can be represented as a set of document semantic entities $E_D = \{e_1, e_2, \cdots, e_p\}$, where each entity, e.g. *paragraph*, *table*, *figure*, also comprised by $e = (text, box)$.

**Task Clarification** Information extraction from VRDs involves fine/coarse-grained processes tailored to the application and the granularity of the information. For the fine-grained level, each token in a sequence $\{t_1, t_2, \cdots, t_n\}$ is classified into predefined categories of the set $\mathbb{Y}$. The goal is to determine the most likely sequence of labels $\{y_1, y_2, \cdots, y_n\}$ corresponding to the token sequence, maximizing $argmax(P(y_1, y_2, \cdots, y_n | t_1, t_2, \cdots, t_n)), y \in Y$. Entity-level extraction, as outlined by Form-NLU (Ding et al., 2023), employs a set of predefined keys $Y_{key_i} \in Y$ and a group of entities $E_D = \{e_1, e_2, \cdots, e_p\}$ to identify and retrieve a specific target entity $e_{k_i}$. This process can be formalized through a model that aims to maximize conditional probability $argmax(P(e_{k_i} | Y_{key_k}, E_D))$.

**Problem Formulation** Suppose $\mathcal{F}$ is a KIE model incorporating pretrained backbones (teachers) from diverse data domains like VRDs (Huang et al., 2022) or natural scene images (Tan & Bansal, 2019), rich in implicit general domain knowledge. $\mathcal{G}$ is an ideal well-trained model in the target domain $\mathbb{D}$, and $\mathcal{D}$ and $\mathcal{L}$ are the probability distance and loss functions, respectively. $\mathcal{F}_t$ is $\mathcal{F}$ trained in the guidance set $\mathbb{D}_g$, represented as $\mathcal{F}_t = argmin(\mathcal{L}(\mathcal{F}(X_{\mathbb{D}_g})))$. $\mathcal{F}_n$ is $\mathcal{F}$ learned on the synthetically annotated dataset $\mathcal{F}_n = argmin(\mathcal{L}(\mathcal{F}(X_{\mathbb{D}_n})))$ and $\mathcal{F}_{nt}$ is $\mathcal{F}_n$ further fine-tuned on $\mathbb{D}_g$, represented as $\mathcal{F}_{nt} = argmin(\mathcal{L}(\mathcal{F}_n(X_{\mathbb{D}_i})))$. Here, $X_{\mathbb{D}}$ denotes the encoded document representation of any target document collection. This paper aims to propose approaches to distill knowledge from pretrained backbones and a synthetically annotated set $\mathbb{D}_n$, to achieve $\mathcal{D}(\mathcal{F}_{nt}, \mathcal{G}) < \mathcal{D}(\mathcal{F}_t, \mathcal{G})$.

## 4 DAViD: DOMAIN ADAPTIVE VISUALLY-RICH DOCUMENT UNDERSTANDING WITH SYNTHETIC INSIGHTS

This section introduces the ***DAViD*** architecture, which consists of two main components: the ***Domain Knowledge Infuser*** and the ***Task-Specific Knowledge Enhancers***. The ***Domain Knowledge Infuser*** is designed to infuse domain-specific knowledge into the model by leveraging synthetic data through various domain adaptation strategies. It is trained on a larger unannotated set $\mathbb{D}_n$, enriched with machine-generated annotations. The ***Task-Specific Knowledge Enhancers*** are responsible for further enhancing the model's performance on specific tasks, utilizing a smaller, well-annotated guidance set $\mathbb{D}_g$. Following the detailed explanation of the DAViD framework, this section outlines the workflow for domain adaptation and task-specific fine-tuning. Additionally, a pseudo-code is provided to guide the implementation of the framework, ensuring clarity and precision in the process.

As demonstrated by previous work (Gu et al., 2021; Ding et al., 2024c), joint-grained document representation learning captures both fine-grained details and coarse-grained relationships, offering a more comprehensive understanding of Visually-Rich documents (Ding et al., 2024a). To this end, we propose the framework $\mathcal{F}$, which is composed of a **Domain Knowledge Infuser** $\mathcal{A}_D$ and two **Task-Specific Knowledge Enhancers**, $\mathcal{A}_T$ and $\mathcal{A}_E$, for refining the model on fine-grained and

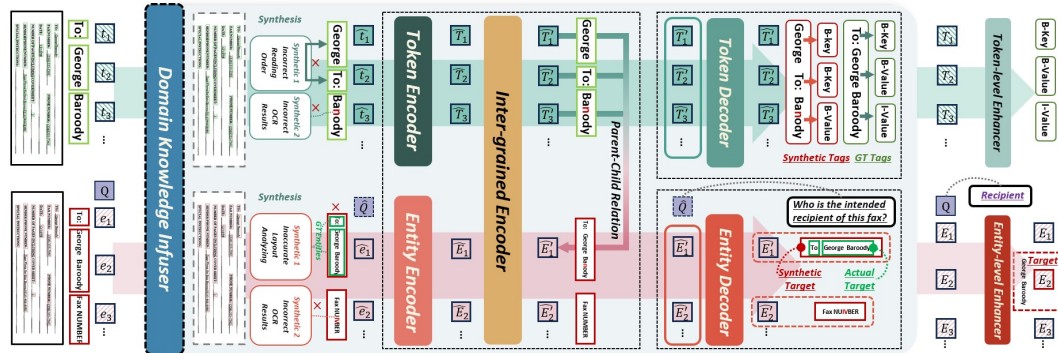

Figure 1: DAViD model architecture contains a Domain Knowledge Infuser and Task-Specific Knowledge Enhancer for various granularity.

coarse-grained tasks, respectively. The Domain Knowledge Infuser $\mathcal{A}_D$ contains **General Domain Encoders (GDEs)** that encode multimodal and multi-grained information from any subset of $\mathbb{D}$. It leverages synthetic data from $\mathbb{D}_n$ to perform various domain adaptation tasks, such as **Structural Domain Shifting (SDS)**, **Synthetic Sequence Tagging (SST)**, and **Synthetic Instructed-Tuning (SIT)**, resulting in the adapted model $\mathcal{A}_{D_n}$. The well-annotated set $\mathbb{D}_g$ is then used to further train $\mathcal{A}_{D_n}$ to refine inter-grained and domain-aware knowledge, and to fine-tune the Task-Specific Knowledge Enhancers $\mathcal{A}_T$ or $\mathcal{A}_E$ for fine-grained or coarse-grained tasks, respectively.

## 4.1 INITIAL REPRESENTATION

For the well-annotated guidance set $\mathbb{D}_g$, each document $D_t \in \mathbb{D}_g$ contains high-quality $n_t$ tokens, represented as $\mathbb{t}_{D_t} = \{t_1, t_2, \ldots, t_{n_t}\}$ and $m_t$ entity annotations, denoted as $\mathbb{e}_{D_t} = \{e_1, e_2, \ldots, e_{m_t}\}$. In contrast, for the unannotated set $\mathbb{D}_n$, we employ standard tools to generate synthetic annotations, resulting in $n_n$ tokens $\mathbb{t}_{D_n} = \{\hat{t}_1, \hat{t}_2, \ldots, \hat{t}_{n_n}\}$ and $m_n$ entities, $\mathbb{e}_{D_n} = \{\hat{e}_1, \hat{e}_2, \ldots, \hat{e}_{m_n}\}$. The tokens can be directly encoded by fine-grained **GDE**, $\mathcal{G}_T$, to obtain fine-grained textual token embedding. For coarse-grained representations, we follow previous work (Luo et al., 2022) by utilizing a pretrained backbone to acquire semantic $S$ and visual $V$ representations of each entity $e$. To better integrate layout information and capture the correlation between token-entity pairs, we introduce a new layout embedding method, named L2V, which converts layout information to visual cues by rendering each input document image to a color-coded image based on the x and y coordinates. A pretrained CNN-backbone extracts RoI features as layout embedding $L$ of $e$ using RoI-Align, similar to visual feature extraction. Thus, each token $t$ and entity $e$ can be represented as $\{t : text, bbox\}$ and $\{e : S, V, L\}$. For any document $D \in \mathbb{D}$, the initial representation of tokens $\mathbb{T}$ and entities $\mathbb{E}$ can be fed into either the token-level general domain encoder $\mathcal{G}_T$ or the entity-level encoder $\mathcal{G}_E$ for comprehensive representation learning.

## 4.2 DOMAIN KNOWLEDGE INFUSER

To acquire the domain-specific knowledge from synthetic document collections in $\mathbb{D}_n$, we introduce the Domain Knowledge Infuser module, $\mathcal{A}_D$, which contains two encoders: $\mathcal{E}_T$ for fine-grained level information and $\mathcal{E}_E$ for coarse-grained information. These encoders serve as the General Domain Encoders(**GDE**s). Various domain knowledge infusion tasks are employed to leverage synthetic annotations and mitigate distribution gaps between general domain pretrained models and target domain $\mathbb{D}$. The following GDE and domain knowledge infusion tasks are designed:

**General Domain Encoding (GDE)** To encode the fine-grained features of any $D \in \mathbb{D}$, we feed the initial token representations $\mathbb{t}$ along with document image $I$ into a VRDU model, $\mathcal{E}_T$, pretrained on a general document collection to obtain a multimodal token representation $\tilde{\mathbb{T}} = \{\tilde{T}_1, \ldots, \tilde{T}_{n'}\}$. Each $\tilde{T}_i$ is additive with the corresponding L2V embedding $L_{T_i}$ to produce the final token representation $T_i$, where all $n'$ tokens in $D$ are represented as $\mathbb{T} = \{T_1, \ldots, T_{n'}\}$. Similarly, the initial entity visual representation $V_j$ of an entity $E_j$ is fed into a visual-language pretrained model (VLPM) $\mathcal{E}_E$, to obtain the augmented $V'_j$. We then fuse multimodal entity representations by linear projection of the concatenated $V'_j$ and $T_j$, addictive with $L_{E_j}$ to get $E_j$, represented as $E_j = Linear(V'_j \oplus T_j) + L_{E_j}$.

All $m'$ semantic entities in document $D$ can be represented as $\mathbb{E} = \{E_1, \ldots, E_{m'}\}$. For coarse-grained level tasks, the query text is fed into $\mathcal{E}_T$ or $\mathcal{E}_E$ to obtain vectorized representations $Q$.

**Structural Domain Shifting (SDS)** To learn the correlation between tokens and entities, we propose a joint-grained transformer encoder, $\mathcal{E}_{jg}$. Document representation learned from general domains are fed into $\mathcal{E}_{jg}$ to obtain augmented token and entity representations, represented as $[\mathbb{T}', \mathbb{E}'] = \mathcal{E}_{ig}([\mathbb{T}, \mathbb{E}])$. To further refine inter-grained contextual learning and acquire more domain-specific knowledge from the large unannotated set $\mathbb{D}_n$, we introduce the inter-grained alignment to predict the existence of parent-child relationships between paired tokens and entities. For any synthetic token-entity pair $(\hat{t}_i, \hat{e}_j)$, where $\hat{t}_i \in \hat{\mathbb{t}}$ and $\hat{e}_j \in \hat{\mathbb{e}}$, we obtain $(\hat{T}_i', \hat{E}_j')$. Then, we compute the alignment score $\gamma$ as:

$$\gamma_{\hat{t}_i, \hat{e}_j} = Linear(\hat{T}_i') \otimes Linear(\hat{E}_i'). \tag{1}$$

If there is a parent-child relation between $\hat{t}_i'$ and $\hat{e}_j'$, then $r_{\hat{t}_i', \hat{e}_j'} = 1$, otherwise $r_{\hat{t}_i', \hat{e}_j'} = 0$. We have a ground truth relation matrix $M_{\hat{\mathbb{t}}, \hat{\mathbb{e}}} = \mathbb{R}^{n' \times m'}$ and a predicted matrix $M_{\hat{\mathbb{t}}, \hat{\mathbb{e}}}'$. The training objective of SDS is to minimize the mean square error between relation matrices:

$$\arg\min_{\theta} \mathcal{L}_{MSE}\left(p(M_{\hat{\mathbb{t}}, \hat{\mathbb{e}}}|\theta), p(M_{\hat{\mathbb{t}}, \hat{\mathbb{e}}}')\right). \tag{2}$$

**Synthetic Sequence Tagging (SST)** To enable the framework to capture fine-grained domain-specific knowledge from $\mathbb{D}_n$, we introduce the synthetic sequence tagging to train the Domain Knowledge Infuser $\mathcal{A}_D$. For a document $D \in \mathbb{D}_n$, each token $\hat{t}_i \in \hat{\mathbb{t}}$ has a corresponding label $\hat{y}_i$, where $\hat{\mathbb{Y}} = \{\hat{y}_1, \ldots, \hat{y}_{n'}\}$. Even if the synthetic labels of $\hat{\mathbb{Y}}$ differ from those in the guidance set $\mathbb{Y}$, training $\mathcal{A}_D$ on SST helps to encode more domain-specific implicit knowledge to enhance fine-grained VRDU tasks. The enhanced token representations $\hat{\mathbb{T}}'$ and entity representations $\hat{\mathbb{E}}'$ are then fed into $\mathcal{D}_T$ as source and memory inputs, refining inter-grained contextual learning. The output $\hat{\mathbb{T}}''$ from $\mathcal{D}_T$ is fed into a linear layer to predict the logits $\hat{\mathbb{Y}}_T'$: $\hat{\mathbb{Y}}_T' = Linear(\mathcal{D}_T(\hat{\mathbb{T}}_T', \hat{E}'))$. The training target is to minimize the cross-entropy loss between $\hat{\mathbb{Y}}'$ and $\hat{\mathbb{Y}}$:

$$\arg\min_{\mathbb{T}''} L_{CE}(p(\hat{\mathbb{Y}}'|\hat{\mathbb{T}}''), p(\hat{\mathbb{Y}})). \tag{3}$$

**Synthetic Instructed-Tuning (SIT)** To enhance the coarse-grained level representations, we introduce a Synthetic Instructed-Tuning, is introduced to train $\mathcal{A}_D$. For each document $D \in \mathbb{D}_n$, we use LLMs to generate synthetic question-answer pairs $\hat{\mathbb{Y}}_E = \hat{Y}_{key_1} : e_{v_1}, \ldots, \hat{Y}_{key_j} : \hat{e}_{v_j}\}$, where $\hat{e}_v \in \hat{E}_{D_t}$. The entity representations are fed as source inputs into entity decoder $\mathcal{D}_E$, with the memory inputs being the combined embedding of synthetic key/question, $\hat{Q}$ and fine-grained representations $\hat{\mathbb{T}}$. A pointer net (PN) is placed on top of linear projection outputs of $\mathcal{D}_E$ to get the final prediction, represented as $\hat{\mathbb{Y}}_E' = PN(Linear(\mathcal{D}_E(\hat{\mathbb{E}}', [\hat{Q}' : \hat{\mathbb{E}}'])))$

### 4.3 Task-Specific Knowledge Enhancers

Task-Specific Knowledge Enhancers are employed to fine-tune the DAViD framework for various downstream tasks using the manually annotated guidance set $\mathbb{D}_g$. The output token embeddings $\mathbb{T}' = \{T_0', \ldots, T_n'\}$ and entity embeddings $\mathbb{E}' = \{E_0', \ldots, E_n'\}$ from Domain Knowledge Infuser $\mathcal{A}_D$ are fed into different Task-Specific Knowledge Enhancers to perform fine-tuning for specific tasks based on the required granularity. For fine-tuning sequence-tagging tasks, a max-pooling layer is applied to extract significant information from each encoding component, which is then fed into a linear classifier:

$$\mathbb{Y}_T' = Linear(Maxpool(\tilde{\mathbb{T}}, \mathbb{T}', \mathbb{T}'')) \tag{4}$$

For coarse-grained entity retrieval tasks, a transformer decoder $\mathcal{D}_{er}$ is used, where the inputs are max-pooled entity representation, and the memory embeddings are the query sequence embeddings:

$$\mathbb{Y}_E' = PN(\mathcal{D}_{er}(Maxpool(\mathbb{E}', \mathbb{E}''), Q)) \tag{5}$$

## 4.4 DOMAIN ADAPTATION AND FINE-TUNING

The entire workflow is systematically outlined to provide clear and reproducible steps for adapting the framework to solve domain-specific document understanding tasks in real-world scenarios. Upon acquiring a domain-intensive document collection $\mathbb{D}$, it is divided into three subsets: $\mathbb{D} = \{\mathbb{D}_n, \mathbb{D}_g, \mathbb{D}_i\}$. Here, $\mathbb{D}_n$ contains the synthetic structure and content information, while $\mathbb{D}_g$ and $\mathbb{D}_i$ are smaller, manually annotated sets used for guidance and practical inference, respectively.

The first stage involves training the Domain Knowledge Infuser on various domain adaptation tasks to generate domain-specific document representations from $\mathbb{D}_n$. Suppose $\hat{\mathbb{T}}$ and $\hat{\mathbb{E}}$ are token and entity representations generated by the GDE. SDS is then conducted to predict parent-child relations between tokens ($\hat{T}'$) and entities ($\hat{E}'$) using the joint-grained encoder $\mathcal{E}_{ig}$. To preserve the joint-grained representation, the pretrained model components will be frozen during the remainder of the domain adaptation and fine-tuning stages. To further enhance fine-grained and coarse-grained representations, two domain adaptation tasks based on synthetic insights are introduced. SST is applied to train the output from $\mathcal{D}_T$, allowing the model to capture more detailed information and utilize preliminary synthetic annotation. Similarly, SIT is used to augment entity representation, making entities query-aware.

---

**Algorithm 1** Overall Workflow

---

**Input:** Specific domain document collection $\mathbb{D}$
**Data Preprocessing:** $\mathbb{D} = \{\mathbb{D}_n, \mathbb{D}_g, \mathbb{D}_i\}$
**Domain Adaptation:** Train $\mathcal{A}_D$ on $\mathbb{D}_n$
i) $GDE(\hat{\mathbb{t}}, \hat{\mathbb{e}}) \xrightarrow{\mathcal{E}_t, \mathcal{E}_e} \hat{\mathbb{T}}, \hat{\mathbb{E}}$
ii) $SDS(\hat{\mathbb{T}}, \hat{\mathbb{E}}) \xrightarrow{\mathcal{E}_{jg}} \hat{\mathbb{T}}', \hat{\mathbb{E}}'$
iii) Freeze $\mathcal{E}_t, \mathcal{E}_e$ and $\mathcal{E}_{jg}$
iv) Fine-grained only: $SST(\hat{\mathbb{t}}, \hat{\mathbb{e}}) \xrightarrow{\mathcal{D}_t} \hat{\mathbb{T}}''$
v) Coarse-grained only: $SIT(\hat{\mathbb{t}}, \hat{\mathbb{e}}) \xrightarrow{\mathcal{D}_e} \hat{\mathbb{E}}'', \hat{Q}''$
**Fine-Tuning:** Train $\mathcal{F}$ on $\mathbb{D}_g$
i) $\mathbb{T}'', \mathbb{E}'' = \mathcal{A}_D(\mathbb{t}, \mathbb{e})$
ii) Fine-grained only: $ST(\mathbb{T}'') \xrightarrow{\mathcal{A}_t} \mathbb{Y}_T$
iii) Coarse-grained only: $ER(\mathbb{E}'', Q'') \xrightarrow{\mathcal{A}_e} \mathbb{Y}_E$
**Inference:** Test $\mathcal{F}$ on $\mathbb{D}_i$

---

After completing the domain-adaptive procedures, the manually annotated tokens $\mathbb{t}_{\mathbb{D}_g}$ and entities $\mathbb{e}_{\mathbb{D}_g}$ are fed into the tuned $\mathcal{A}_D$ to obtain $\mathbb{T}_{\mathbb{D}t}$ and $\mathbb{E}_{\mathbb{D}_g}$. These representations are then fine-tuned using Task-Specific Knowledge Enhancers. The final framework is evaluated on the inference set $\mathbb{D}_i$.

## 5 ENVIRONMENTAL SETUP

### 5.1 BENCHMARK DATASETS

We evaluate our proposed DAViD framework on two benchmark datasets to demonstrate its ability to capture domain-specific layouts and semantic information from documents enriched with synthetic insights. The selected datasets simulate real-world scenarios where the framework must adapt to diverse document formats and content complexities: **1) CORD** (Park et al., 2019) includes 800 training, 100 validation, and 100 test samples with multi-level annotations for printed/scanned ($\mathcal{P}$) receipt understanding. In line with previous document understanding frameworks (Xu et al., 2021; Huang et al., 2022), we focus on sequence tagging (ST) to identify entity types like "*store name*", "*menu quantity*", and "*void total* **2) Form-NLU** (Ding et al., 2023) contains 535 training, 76 validation, and a test set with 50 printed (P) and 50 handwritten (H) samples. From the dataset, we focus on particularly Task B, which involves extracting key information from digital ($\mathcal{D}$), printed ($\mathcal{P}$), and handwritten ($\mathcal{H}$) forms. This task provides ground truth bounding boxes for form semantic entities (e.g., "*Shareholder Name*", "*Share Class*") to facilitate target entity retrieval.

To prepare the benchmark datasets, as shown by Figure 2, we apply *a) Document Collection Re-allocation* and *b) Synthetic Layout Annotation* for structural adaptation across all datasets. For task-specific knowledge enhancers, additional procedures like *c) Synthetic Sequence Tagging* and *d) Synthetic Inquiry Generation* simulate practical scenarios, enabling DAViD to capture domain-specific variations and semantic relationships. *a) Document Collection Re-allocation* replicates real-world scenarios by dividing the original dataset into three subsets: synthetic annotated, manually annotated, and test sets. The original training set is used as the synthetic annotated set, the validation set as the fully annotated set, and the test set for evaluation. Synthetic annotations are generated using off-the-shelf tools to help the model learn and differentiate between layout and semantic information at various granularities. *b) Synthetic Layout Annotation* extracts grouped textual tokens, textlines, or document semantic entities to capture layout structures. Tools like PDFMiner, OCR tools [1], and

---

[1] For example, PaddleOCR: `https://github.com/PaddlePaddle/PaddleOCR` is widely used.

Figure 2: Workflow for generating synthetic annotations for domain-specific understanding.

layout analysis models generate synthetic layout annotations, capturing bounding box coordinates and textual content. *c) Synthetic Sequence Tagging* creates synthetic annotations for token sequences to support fine-grained sequence tagging. Large language models (LLMs) generate labels for each document, which may differ from manually annotated labels. Fine-tuning these synthetic annotations enhances the model's contextual understanding. *d) Synthetic Inquiry Generation* uses question-answer pairs generated by LLMs to leverage general textual knowledge. Prompts are designed to extract QA pairs, then matched with entities from layout analyzers. The highest-matched entity serves as the retrieval target for instructed tuning [2].

## 5.2 BASELINES AND IMPLEMENTATION DETAILS

We employ a variety of pretrained backbones from both fine-grained and entity-level frameworks to encode multi-granularity features. To evaluate the effectiveness of existing LLMs/MLLMs on VRDU tasks, different models are tested under zero-shot settings[3]. **1) Fine-grained Baselines** We utilize three recently proposed fine-grained document understanding models: LayoutLMv3 (Huang et al., 2022), LiLT (Wang et al., 2022), and UDop (Tang et al., 2023), which leverage multimodal information pretrained on general document collections, like IIT-CDIP (Lewis et al., 2006), to perform key information extraction through sequence tagging tasks, achieving state-of-the-art performance when fully trained on benchmark datasets. **2) Entity-level Baselines** For entity-level document understanding, we include RoI-based Vision-Language Pretrained Models (VLPMs) such as LXMERT (Tan & Bansal, 2019) and VisualBERT (Li et al., 2019) as baselines for entity retrieval. Initially pretrained on natural scene images, these models are further adapted through transfer learning and domain-specific knowledge infusion, enabling effective key information extraction and question answering in Visually-Rich Documents (VRDs). **3) Zero-shot LLMs and MLLMs** LLMs and MLLMs have shown impressive zero-shot performance across diverse domains. To assess their capabilities on VRDU tasks, we evaluate GPT-3.5 and GPT-4, leading closed-source models for mono-modality and multimodal tasks. For open-source models, we select QWen-VL (Bai et al., 2023) (pretraining-based), LLAVA-1.5 (Liu et al., 2024) (instruct-tuned), and BLIP-3 (Xue et al., 2024) (pretrained with instruct-tuning) based on their distinct training strategies. We follow the configurations of baseline models for both token and entity levels as specified in (Huang et al., 2022; Wang et al., 2022; Tang et al., 2023; Ding et al., 2023). Implementation Details are in Appendix B.

## 6 RESULTS AND DISCUSSION

We conduct comprehensive experiments accompanied by an in-depth analysis to demonstrate the effectiveness of the proposed frameworks across diverse scenarios. Furthermore, additional robustness evaluations, along with the impact analysis of varying quantities of the synthetic dataset, are provided in Appendix D for a more thorough comparison and understanding.

### 6.1 PERFORMANCE ANALYSIS

**Overall Trend** Table 1 presents the performance of various model configurations, demonstrating the effectiveness of the proposed domain adaptation methods in capturing domain knowledge. Due to their strong baseline performance, LayoutLMv3 and LXMERT were selected as token and entity

---

[2]More detailed dataset statistics and synthetic data analysis please refer to Appendix C

[3]Please refer to Appendix A to check more details about each group of models.

encoders to construct the joint-grained Domain Knowledge Infusers $\mathcal{A}_D$ within the framework $\mathcal{F}$. The results show that integrating fine and coarse-grained information within $\mathcal{F}$ outperforms mono-grained baselines, boosting downstream task performance. We note that incorporating fine-grained features significantly enhanced entity representation in FormNLU, with a performance gain of approximately 8% for the printed and 21% for the handwritten sets. All domain adaptation methods, including the novel L2V positional features, improved performance. Detailed analyzes are in subsequent sections.

**Breakdown Analysis** Table 2 compares performance across various information categories, highlighting the benefits of the joint-grained framework in generating comprehensive representations. This framework enriches entity semantics and token structures, leading to notable improvements—such as a 58% increase in "*cid*" in FormNLU-H and an 18% increase in "*SubC*" in CORD. While L2V enhances feature representation overall, it may introduce inconsistencies in flexible layout categories, like handwritten "*cid*" in FormNLU. The proposed methods, especially SDS, consistently show robust improvements across most categories, demonstrating their effectiveness in capturing domain-aware knowledge. Although leveraging LLM-generated tags (SST) or QA pairs (SIT) boosts performance, it may lead to occasional instability. For example, combining SDS with SST or SIT improve specific categories but may yield lower results in others—such as a 20% decrease in CORD's "*SubC*" when using SDS+SST compared to SST.

| Entity Level | FormNLU | | Token Level | CORD |
|---|---|---|---|---|
| | P | H | | |
| **Full Training Set** | | | | |
| Transformer | 88.62 | 74.06 | LayoutLMv3 | 96.56 |
| VisualBERT | 85.90 | 70.14 | LiLT | 96.07 |
| LXMERT | 94.15 | 82.80 | UDOP | 97.58 |
| **Tuning in Guidance Set ($\mathbb{D}_g$)** | | | | |
| Transformer | 72.82 | 60.30 | LayoutLMv3 | 87.08 |
| VisualBERT | 46.48 | 48.41 | LiLT | 86.74 |
| LXMERT | 81.21 | 64.66 | UDOP | 80.88 |
| Joint-grained | 89.60 | 85.76 | Joint-grained | 87.48 |
| + L2V | 90.60 | 87.60 | + L2V | 88.11 |
| + SDS | 91.11 | **88.78** | + SDS | 89.08 |
| + SIT | 90.77 | 87.94 | + SST | 88.83 |
| + SIT + SDS | **92.62** | 88.61 | + SST + SDS | **90.25** |

Table 1: Performance using full and limited training sets with domain adaptation strategies.

| Entity Level | FormNLU | | | | | | | | Token Level | CORD | | | |
|---|---|---|---|---|---|---|---|---|---|---|---|---|---|
| | *cid* | | *pdt* | | *gdt* | | *pvp* | | | *SubC* | *UP* | *CCP* | *SubO* |
| | P | H | P | H | P | H | P | H | | | | | |
| LXMERT | 45.83 | 30.00 | 72.00 | 69.39 | 78.00 | 83.67 | 98.00 | 67.35 | LayoutLMv3 | 55.17 | 93.53 | 85.71 | 82.54 |
| Joint-grained | 50.00 | **88.00** | 66.00 | 18.37 | 92.00 | 79.80 | **100.00** | 89.80 | Joint-grained | 73.33 | 85.51 | 91.67 | 76.92 |
| + L2V | 66.67 | 72.00 | 72.00 | 61.22 | 88.00 | **95.92** | **100.00** | 95.92 | + L2V | 64.29 | 94.12 | 84.62 | 82.54 |
| + SDS | **79.17** | **88.00** | 66.00 | 61.22 | 88.00 | 89.80 | **100.00** | 95.92 | + SDS | 80.00 | 94.89 | **100.00** | 89.23 |
| + FST | 62.50 | 78.00 | 72.00 | 67.35 | 90.00 | **100.00** | **100.00** | **100.00** | + SST | **84.85** | 91.43 | 80.00 | 80.65 |
| + FST + SDS | **79.17** | 78.00 | **80.00** | **81.63** | 92.00 | 85.71 | 96.00 | 95.92 | + SST + SDS | 64.29 | **97.06** | 88.89 | **90.32** |

Note: '*cid*' = Company ID (ACN/ARSN), '*pdt*' = Previous Notice Date '*gdt*' = Given Date, '*pvp*' = Previous Voting Power
'*SubC*' = Subtotal Count, '*UP*' = Unit Price, '*CCP*' = Credit Card Price, '*SubO*' = subtotal others

Table 2: Selective breakdown results of performance across representative categories.

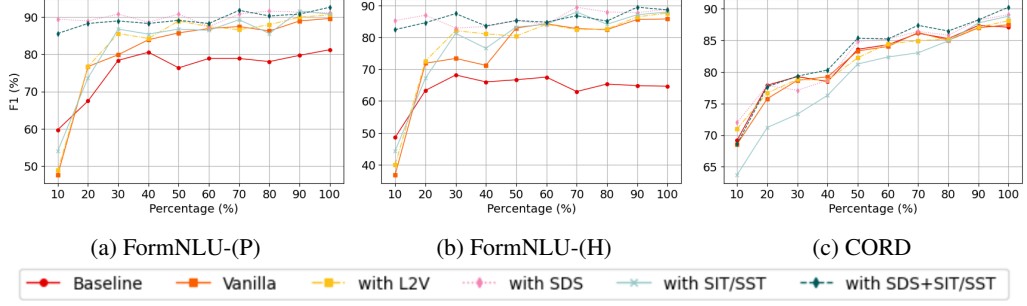

(a) FormNLU-(P)      (b) FormNLU-(H)      (c) CORD

Figure 3: Performance of our model with stepped training set ratios on three test sets.

## 6.2 RESULTS OF FINE-TUNING WITH VARYING TRAINING RATIOS

**Few-shot Testing** We evaluated the robustness of our methods with varying amounts of annotated data from $\mathbb{D}_g$, using training sizes from 10% to 100% of $\mathcal{D}_t$. As shown in Table 1, domain adaptation consistently outperformed non-adapted baselines by leveraging domain-specific information from the synthetic dataset $\mathbb{D}_n$, although performance sensitivity varied across different tasks and training sizes. For the entity-level FormNLU, both printed (P) and handwritten (H) test sets improved as training sizes increased. Without domain adaptation, performance was poor in few-shot scenarios. With just 10% of $\mathbb{D}_g$, SDS achieved over 80% accuracy on both P and H sets, demonstrating its

ability to capture domain-specific structural information and enhance semantic understanding. For token-level results in CORD, incorporating coarse-grained information improved performance across training sizes. SDS consistently outperformed other configurations, effectively utilizing synthetic structural information from $\mathbb{D}_n$. However, SIT and SST underperformed in few-shot settings, likely due to reliance on synthetic LLM-generated samples that need more data to bridge distribution gaps.

**Zero-shot Testing** We evaluated zero-shot performance (Table 3) to assess domain knowledge capture. SDS effectively distilled structural knowledge from $\mathbb{D}_n$, achieving 87.42% on FormNLU (printed) and 81.74% (handwritten). In contrast, SIT showed minor improvements on the printed set but decreased on the handwritten set, due to the distribution gap between digital-born QA pairs from $\mathbb{D}_g$ and handwritten tests. For CORD, domain adaptation shows less impact than entity-level tasks, as the joint-grained framework benefits entity representations more than fine-grained token representations. Entities can contextually learn from tokens, improving semantic understanding and attention alignment during domain adaptation and fine-tuning. Tokens gain less from coarse-grained embeddings, highlighting the need for joint-grained frameworks as a future research direction.

| FormNLU | | | CORD | |
|---|---|---|---|---|
| Config | P | H | Config | Test |
| Baseline | 1.67 | 0.5 | Baseline | 0 |
| Joint-grained | 0 | 0 | Joint-grained | 0 |
| + L2V | 0 | 0 | + L2V | 0 |
| + SDS | **87.42** | **81.74** | + SDS | 0.05 |
| + SIT | 5.7 | 0.17 | + SST | 0.25 |
| + SIT + SDS | 47.65 | 44.22 | + SST + SDS | **4.21** |

Table 3: Comparison of zero-shot performance on various configurations.

## 6.3 Ablation Study

**Effects of Training Epochs** We observed that varying the number of training epochs (with ep. 1 representing one epoch in Table 4) for different domain adaptation methods impacts fine-tuning results. Insufficient training can result in limited domain-specific information capture. For instance, training the SDS+SST method for just one epoch on the CORD dataset yields about 2.5% lower performance than two epochs. Conversely, increasing training epochs can cause the model distribution to shift closer to $\mathbb{D}_n$, but further away from $\mathbb{D}_g$. For example, training SDS+SIT for three epochs on the FUNSD dataset resulted in a performance drop of approximately 2.5% and 5% on sets P and R, respectively. Finding the optimal number of epochs for each domain adaptation strategy requires careful adjustment based on the specific dataset and task.

**Effects of Freezing** To retain domain knowledge infused from $\mathbb{D}n$ by the joint-grained encoder $\mathcal{E}_{jg}$, freezing its parameters after applying SDS proved beneficial. It preserved the learned structure and semantic insights, leading to better performance during fine-tuning. As shown in Table 4, unfreezing the models resulted in lower performance. For example, SDS+SIT on FormNLU-P dropped from 92.62% to 88.58% when the parameters were not frozen.

**Effects of L2V** We evaluated the impact of the L2V positional feature on domain adaptation methods. As shown in Table 4, removing L2V led to an approximate 2% performance drop. This suggests that L2V enhances positional-awareness in token and entity representations, contributing to better document understanding.

| FormNLU | | | CORD | |
|---|---|---|---|---|
| Config | P | H | Config | Test |
| SDS (ep. 1) | 91.11 | 88.78 | SDS (ep. 1) | 88.45 |
| SDS (ep. 2) | 89.93 | 86.60 | SDS (ep. 2) | 89.08 |
| SDS (ep. 3) | 91.11 | 84.42 | SDS (ep. 3) | 87.35 |
| SIT (ep. 1) | 90.94 | 87.77 | SST (ep. 1) | 88.83 |
| SIT (ep. 2) | 86.91 | 83.75 | SST (ep. 2) | 87.54 |
| SIT (ep. 3) | 86.07 | 81.41 | SST (ep. 3) | 85.71 |
| SDS+SIT (ep. 1) | 91.11 | 89.11 | SDS+SST (ep. 1) | 86.95 |
| SDS+SIT (ep. 2) | 92.62 | 88.61 | SDS+SST (ep. 2) | 90.25 |
| SDS+SIT (ep. 3) | 87.58 | 83.92 | SDS+SST (ep. 3) | 87.49 |
| SDS Frozen | 91.11 | 88.78 | SDS Frozen | 89.08 |
| SDS Unfrozen | 91.61 | 85.59 | SDS Unfrozen | 86.91 |
| SDS+SIT Frozen | 92.62 | 85.59 | SDS+SST Frozen | 90.25 |
| SDS+SIT Unfrozen | 88.59 | 85.93 | SDS+SST Unfrozen | 86.64 |
| SDS with L2V | 91.11 | 89.11 | SDS with L2V | 89.08 |
| SDS without L2V | 89.26 | 84.25 | SDS without L2V | 87.57 |
| SIT with L2V | 90.94 | 87.77 | SST with L2V | 88.83 |
| SIT without L2V | 85.91 | 87.94 | SST without L2V | 87.19 |

Table 4: Ablation results for FormNLU and CORD

## 6.4 Comparison with LLMs/MLLMs

We evaluated the state-of-the-art LLMs and MLLMs to address VRDU tasks using various mono- and multi-modal prompts across different model checkpoints based on various training approaches, comparing their performance and efficiency with the DAViD framework in Table 5. For close-source GPT-4o, two prompts were used: the text-only prompt $P_t : \{K, C\}$, where $K$ is the key text content and $C$ is the provided text content, and the text-vision prompt $P_{tv} : \{K, C, I\}$, where $I$ is the target

form image. GPT-3.5 uses $P_t$ only and other open source MLLMs are used $P_{tv}$ to leverage text and vision information. GPT-4o with prompt $P_t$ outperforms GPT-3.5 using the same prompt, while with the multimodal prompt $P_{tv}$, GPT-4o achieves around a 13% increase in F1 score. Other open-source MLLMs show an apparent gap between close GPT-series [4].

However, a significant gap remains between the results of DAViD tuned on the guidance set $\mathbb{D}_g$ and even the zero-shot setting DAViD-ZS. LLMs/MLLMs still struggle with VRDU under zero-shot scenarios, especially open-source MLLMs. In contrast, the DAViD demonstrates superior performance, suggesting that the proposed frameworks and domain adaptation techniques effectively distil knowledge from both LLMs and VLPMs. Furthermore, the performance of DAViD could be further enhanced by improving the quality of the synthetically annotated set $\mathbb{D}_n$ and incorporating more representative backbone architectures. We evaluated that of LLMs and MLLMs on a subset of the CORD

| Model | FormNLU P | | FormNLU H | | CORD* | |
|---|---|---|---|---|---|---|
| | Time | F1 | Time | F1 | Time | ANLS |
| GPT-3.5 | 03:49 | 34.37 | 04:38 | 30.94 | 01:16 | 28.15* |
| GPT-4o ($P_t$) | 04:46 | 42.09 | 04:19 | 36.00 | 01:48 | 29.55* |
| LLava ($P_{tv}$) | 52:54 | 9.79 | 60:58 | 7.82 | 10:23 | 37.98 |
| QWen ($P_{tv}$) | 1:36:00 | 9.84 | 1:58:00 | 8.43 | 18:13 | 37.58 |
| Blip3 ($P_{tv}$) | 36:06 | 12.62 | 35:24 | 11.67 | 10:12 | 43.73 |
| GPT-4o ($P_{tv}$) | 20:02 | 59.88 | 20:49 | 49.15 | 07:55 | 79.46* |
| DAViD-ZS | 03:37 | 87.42 | 03:31 | 81.74 | - | - |
| DAViD-$\mathbb{D}_g$ | 03:37 | **92.62** | 03:31 | **88.78** | 00:31 | **90.25** |

Table 5: Performance between LLM/MLLMs and DAViD. CORD* is adopted QA-style subset introduced by LayoutLLM.

dataset provided by LayoutLLM (Luo et al., 2024), and the results indicate that the performance of LLMs/MLLMs remains suboptimal for this task, as well as with less efficiency.

# 7 QUALITATIVE ANALYSIS: CASE STUDIES

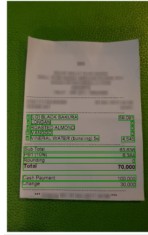 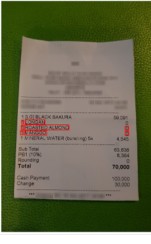 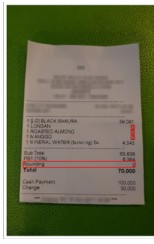 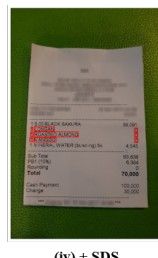 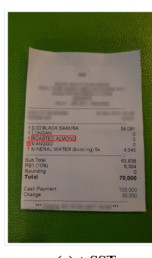 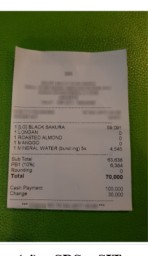

(i) Ground Truth  (ii) LayoutLMv3  (iii) Joint-grained  (iv) + SDS  (v) + SST  (vi) + SDS + SIT

Figure 4: Real-world CORD dataset sample: (i) Ground truth key information highlighted in green. (ii) - (v) Incorrect predictions marked with red rectangles under various configurations. (vi) The best performance was achieved using two domain adaptation methods, with no incorrect predictions.

To qualitatively demonstrate the effectiveness of the proposed framework, a real-world example from the CORD is presented in Figure 15. Compared to baseline models, the joint-grained framework produces fewer incorrect predictions, likely due to the integration of coarse-grained information. In this case, while SDS alone does not improve results, the SST approach shows noticeable enhancements. Furthermore, combining both domain adaptation methods results in entirely accurate predictions. This highlights the effectiveness of proposed domain adaptation techniques in leveraging domain knowledge from noisily annotated data to improve downstream task performance [5].

# 8 CONCLUSION

This paper presents DAViD, a framework that enhances VRDU by capturing domain-specific knowledge using synthetic annotations, achieving strong performance with minimal labeled data. DAViD utilizes domain adaptation techniques to transition from general-purpose encoders to those optimized for domain-specific document collections. The framework introduces SDS to create a robust joint-grained representation by aligning fine- and coarse-grained features. For granularity-specific tasks, LLMs generate synthetic annotations, supporting SIT and SST. Extensive evaluations demonstrate that DAViD effectively captures domain-specific knowledge, significantly improving performance and robustness across benchmarks with limited annotated samples.

---

[4]Appendix D.5.1 provides prompt details. Detailed LLM-based analysis are in Appendix D.5.2and D.6
[5]More visualized quantitative examples with analysis could be found in Appendix E.2

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

## A BASELINE MODELS

### A.1 FINE-GRAINED DOCUMENT UNDERSTANDING FRAMEWORKS

- **LayoutLM-v3** (Huang et al., 2022): is the first model to leverage visual cues in VRDU without using pretrained CNN backbones. Various pretraining methods were proposed to fuse the multimodal features from the general domain and achieve SOTA on several VRDU downstream tasks.

- **LiLT** (Wang et al., 2022): is a language-independent layout transformer which supports pertained on a single language document collections but fine-tuned on other language tasks. A bi-directional attention complementation mechanism to learn the layout and textual modality interaction with layout-aware pretraining tasks for capturing more general document text-layout interaction.

- **UDop** (Tang et al., 2023): is an encoder-decoder structure that leverages text, image and layout modalities to conduct the VRDU tasks in a sequence generation style. UDop is pretrained in a cross-modal, self-supervised learning way and pretrained supervised tasks on cross-domain benchmark datasets to acquire more robust representations.

### A.2 COARSE-GRAINED VISION-LANGUAGE PRETRAINED MODELS

- **VisualBERT** (Li et al., 2019): is a transformer-based vision-language pretrained (VLPM) model that contextualizes the understanding of visual cues from detected regions of interest (RoI) and accompanying text within the domain of general scene images.

- **LXMERT** (Tan & Bansal, 2019): is a VLPM that utilizes the bounding boxes of Regions of Interest (RoIs) to capture spatial relations between them. This approach leads to a more comprehensive multimodal representation for general domain vision-language tasks.

### A.3 LLMS/MLLMS FOR ZERO-SHOT TESTING

- **LLaVA-1.5** (Liu et al., 2024): is built upon LLaVA, which was the first model to extend instruction-tuning to the language-image multimodal space. LLaVA-1.5 addresses LLaVA's limitations, particularly its underperformance in generating short-form answers on academic benchmarks, by introducing a new MLP-based cross-modal connector and employing scaling-up techniques, such as handling high-resolution images. We use `llava-hf/llava-1.5-7b-hf` checkpoints for zero-shot testing.

- **QWen-VL** (Bai et al., 2023): QWen-VL employs the large language model QWen-7B as its foundational component and integrates a Vision Transformer as the vision encoder. These components are jointly trained using a cross-attention-based vision-language adaptor. The model undergoes a two-stage pretraining process, initially learning from large-scale weakly labeled image-text pairs, followed by fine-tuning with high-quality, fine-grained vision-language annotations. We use `Qwen/Qwen-VL` checkpoints for zero-shot testing.

- **xGen-MM** (Xue et al., 2024): adopts ViT as its vision encoder, incorporating a perceiver resampler to downsample the image embeddings, with phi3-mini serving as the large language model decoder. This framework is designed to scale up LLM training by leveraging a combination of multimodal interleaved datasets, curated caption datasets, and other publicly available sources. We use `Salesforce/xgen-mm-phi3-mini-instruct-r-v1` checkpoints for zero-shot testing.

- **GPT-3.5** (OpenAI, 2023): is one of the most powerful closed-source mono-modality LLMs, achieving remarkable performance and being widely employed across diverse daily applications such as customer support, content creation, and language translation. It is frequently used as a baseline for evaluating zero-shot performance on linguistic-related tasks. We use `gpt-3.5-turbo-0125` checkpoints for zero-shot testing.

- **GPT-4o** (OpenAI, 2024): is an advanced multimodal LLM that extends its capabilities to process diverse inputs, including language, vision, and audio. It demonstrates exceptional performance across various multimodal benchmark datasets and is widely used as

a baseline for assessing zero-shot performance in complex multimodal tasks. We use `gpt-4o-2024-08-06` checkpoints for zero-shot testing.

## B  IMPLEMENTATION DETAILS

We follow the configurations of baseline models for both token and entity levels as specified in (Huang et al., 2022; Wang et al., 2022; Tang et al., 2023; Ding et al., 2023). LayoutLMv3 and LXMERT are used as the token ($\mathcal{E}_T$) and entity ($\mathcal{E}_E$) encoders, respectively, based on their proven performance. Our architecture features six-layer transformer encoders with a hidden size of 768 for the joint-grained encoder ($\mathcal{E}jg$). Two additional six-layer transformer decoders with a hidden size of 768 serve as the token ($\mathcal{D}_T$) and entity ($\mathcal{D}_E$) decoders. We maintain a consistent learning rate of 2e-5 and a batch size of 2 for domain adaptation and fine-tuning phases. All experiments are conducted on a 16GB NVIDIA V100 GPU, with 60 epochs for CORD and 15 for Form-NLU. For open source LLMs/MLLMs, all zero-shot experiments are conducted on a 22.5GB NVIDIA L4 GPU.

## C  DATASET INFORMATION

### C.1  DATASET STATISTICS

The detailed statistics of adopted datasets with the machine-generated synthetic set statistics are listed there. For FormNLU datasets, as it's a text-embedded form that can be processed by the PDF parser, the number of entities is counted as the textlines extracted by the PDFMiner. For the CORD dataset, we use PaddleOCR to extract the text lines of the scanned receipts to acquire 13,200 entities.

| Dataset | Split | | | Year | Domain | Task | Script | Lang. | Synthetic Dataset Size | | | |
|---------|-------|-----|------|------|--------|------|--------|-------|--------|----------|------|-------|
| | Train | Val | Test | | | | | | # IMG | # Entities | # QA | # Cat |
| FormNLU | 535 | 76 | 50/50 | 2023 | Financial Form | Key Entity Retrieval | P/H | English | 535 | 103866 | 15278 | N/A |
| CORD | 800 | 100 | 100 | 2019 | Receipt | Sequence Tagging | P | English | 800 | 13200 | N/A | 40 |

Table 6: Original and synthetic annotated datasets of adopted datasets.

### C.2  SYNTHETIC DATA ANALYSIS

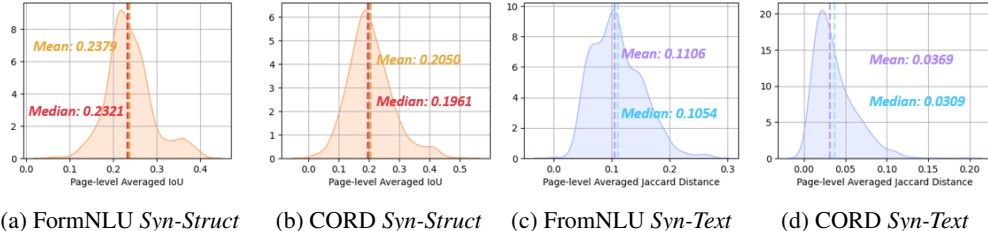

(a) FormNLU *Syn-Struct*   (b) CORD *Syn-Struct*   (c) FromNLU *Syn-Text*   (d) CORD *Syn-Text*

Figure 5: Off-the-shelf-tool analysis. Synthetic-Structure (*Syn-Struct*) and Synthetic-Text (*Syn-Text*).

We analyze the distribution characteristics of synthetic annotations generated by off-the-shelf tools, focusing on two primary types: **1) Layout structure variations** arise from inaccuracies in the regions of document semantic entities extracted by document parsing tools. However, text content variations result from improperly grouped words and misrecognized text during the parsing process. From Figures 5b and 5a, most documents exhibit mismatches in layout structures, with the average Intersection over Union (IoU) between detected entities and ground truth entities falling below 0.3 in both datasets. **2) Text content variations** exhibit even lower Jaccard similarities, dropping below 0.2 for Form-NLU and 0.1 for CORD. Errors in entity detection can propagate during text recognition, resulting in a larger distribution gap between extracted text sequences and the ground truth. Compared to text-embedded source files that can be processed by PDF parsing tools like PDFMiner, scanned documents processed by OCR tools tend to introduce even more variations, further complicating the adaptation of models to these documents.

# D ADDITIONAL EVALUATION RESULTS

## D.1 ALL BREAKDOWN RESULTS

In Section 6.1 of the main paper, we analyze the performance under different configurations of selective categories. This section presents detailed experimental results for each sub-category, providing insights into the effects of the proposed methods and modules on specific categories.

### D.1.1 FORMNLU DATASET

Tables 7 and 8 compare the performance of the printed and handwritten sets. Overall, the printed set demonstrates better performance, particularly for target entities located in the "*Table*" area. This may be due to a smaller domain gap between the digital training set and the printed set P, compared to the handwritten set H. Additionally, joint-grained frameworks consistently outperform mono-grained baselines, and incorporating domain adaptation methods significantly enhances both performance and robustness across the framework.

| Model | F1 | cnm | cid | hnm | hid | cdt | pdt | gdt | cls | ppn | pvp | cpn | cvp |
|---|---|---|---|---|---|---|---|---|---|---|---|---|---|
| LXMERT | 81.21 | 94.00 | 84.00 | 79.17 | 45.83 | 78.00 | 72.00 | 78.00 | 72.00 | 94.00 | 98.00 | 82.00 | 96.00 |
| Joint-grained | 89.60 | 98.00 | 92.00 | **97.92** | 50.00 | 88.00 | 66.00 | **92.00** | 100.00 | 100.00 | 100.00 | 92.00 | **98.00** |
| + L2V | 90.60 | 98.00 | **98.00** | 79.17 | 66.67 | **94.00** | 72.00 | 88.00 | 98.00 | 98.00 | 100.00 | 96.00 | **98.00** |
| + SDS | 91.11 | **100.00** | 94.00 | 91.67 | 79.17 | 90.00 | 66.00 | 88.00 | 100.00 | 86.00 | 100.00 | 100.00 | **98.00** |
| + SIT | 90.77 | 96.00 | 94.00 | 93.75 | 62.50 | 82.00 | 72.00 | 90.00 | 100.00 | 100.00 | 100.00 | 100.00 | **98.00** |
| + SIT + SDS | **92.28** | 98.00 | 94.00 | 95.83 | **79.17** | 86.00 | **80.00** | **92.00** | 98.00 | 92.00 | 96.00 | 98.00 | **98.00** |

Table 7: Model breakdown performance on FormNLU printed set. Explanation of abbreviations: cnm (Company Name/Scheme), cid (Company ID), hnm (Holder Name), hid (Holder ID), cdt (Change Date), pdt (Previous Notice Date), gdt (Given Date), cls (Class of Securities), ppn (Previous Person's Votes), pvp (Previous Voting Power), cpn (Current Person's Votes), cvp (Current Voting Power).

| Model | F1 | cnm | cid | hnm | hid | cdt | pdt | gdt | cls | ppn | pvp | cpn | cvp |
|---|---|---|---|---|---|---|---|---|---|---|---|---|---|
| LXMERT | 64.66 | 66.00 | 76.00 | 88.00 | 30.00 | 58.00 | **69.39** | **83.67** | 8.00 | 84.00 | 67.35 | 72.00 | 74.00 |
| Joint-grained | 85.76 | **100** | **100** | **100** | **88.00** | 92.00 | 18.37 | 79.59 | **94.00** | 90.00 | 89.80 | **90.00** | **96.00** |
| + L2V | 87.60 | **100** | 98.00 | 96.00 | 72.00 | **96.00** | 61.22 | **95.92** | **100** | 92.00 | **95.92** | 62.00 | 92.00 |
| + SDS | **88.78** | **100** | **100** | **100** | **88.00** | 92.00 | 61.22 | 89.80 | 84.00 | 88.00 | **95.92** | 82.00 | 84.00 |
| + SIT | 87.94 | **100** | 98.00 | **100** | 78.00 | 60.00 | 67.35 | 85.71 | **100** | 98.00 | 100.00 | 88.00 | 80.00 |
| + SIT + SDS | 88.61 | **100** | 96.00 | 98.00 | 78.00 | 78.00 | **81.63** | 85.71 | 86.00 | 92.00 | **95.92** | 90.00 | 82.00 |

Table 8: Model breakdown performance on FormNLU handwritten set. Explanation of abbreviations: cnm (Company Name/Scheme), cid (Company ID), hnm (Holder Name), hid (Holder ID), cdt (Change Date), pdt (Previous Notice Date), gdt (Given Date), cls (Class of Securities), ppn (Previous Person's Votes), pvp (Previous Voting Power), cpn (Current Person's Votes), cvp (Current Voting Power).

### D.1.2 CORD DATASET

The overall and breakdown results of CORD datasets are also represented in Table 9 and 10. Compared with integrating fine-grained level information to coarse-grained, there is limited improvement on integrating coarse-grained information to fine-grained baselines.

## D.2 STEPPED GUIDANCE SET RATIO RESULTS

To explore the effects of the size of the guidance set on test set performance, we reported and analyzed the performance in Figure 3. The exact performance of each guidance set ratio is listed in an additional analysis.

### D.2.1 FORMNLU DATASET

In the FormNLU dataset, both the printed set (P) and handwritten set (H) exhibit similar patterns as represented by Table 11 and Table 12. While incorporating fine-grained information can enhance

| Model | Overall | CNT | DscP | NM | Num | Prc | SubC | SubNM | SubPrc | UP | CshPrc |
|---|---|---|---|---|---|---|---|---|---|---|---|
| LayoutLMv3 | 87.08 | 96.00 | 47.06 | 92.80 | 58.82 | 93.59 | 55.17 | 55.56 | 50.00 | 93.53 | **66.67** |
| Joint-grained | 87.48 | 96.02 | 47.06 | 92.87 | **76.19** | 93.15 | 73.33 | 57.53 | 72.73 | 85.51 | 46.15 |
| + L2V | 88.11 | 95.81 | 44.44 | 91.60 | 62.50 | 94.35 | 64.29 | 57.14 | 58.82 | 94.12 | 62.50 |
| + SDS | 89.08 | **97.53** | 44.44 | 92.57 | 30.77 | 95.09 | 80.00 | **62.16** | 64.86 | 94.89 | 55.56 |
| + SST | 88.83 | 95.59 | **58.33** | **93.26** | 58.82 | 93.93 | **84.85** | **62.16** | 60.00 | 91.43 | 62.50 |
| + SST + SDS | **90.25** | 95.59 | 53.33 | 92.08 | 73.68 | **95.48** | 64.29 | 52.46 | **74.29** | **97.06** | 50.00 |

Table 9: Model Comparison on Various Metrics (Part 1), including count (CNT), discount price (DscP), miscellaneous items (Etc), item subtotal (ItmSubT), name (NM), number (Num), price (Prc), subtotal count (SubC), sub name (SubNM), subtotal price (SubPrc), and unit price (UP).

| Model | ChgPrc | CCP | EMP | MQtyC | MTypC | TotEtc | TotPrc | DscPrc | SubO | SrvPrc | STP |
|---|---|---|---|---|---|---|---|---|---|---|---|
| LayoutLMv3 | 13.33 | 85.71 | 87.94 | 89.13 | 84.14 | 83.72 | 58.54 | 40.00 | 82.54 | 16.67 | 18.18 |
| Joint-grained | 0.00 | 91.67 | 91.55 | 86.87 | 86.30 | 94.12 | 50.91 | 28.57 | 76.92 | 36.36 | 0.00 |
| + L2V | 0.00 | 84.62 | **92.65** | **93.62** | 87.42 | 94.02 | 57.14 | 16.67 | 82.54 | 20.00 | **28.57** |
| + SDS | 0.00 | **100.00** | 90.65 | 91.49 | **92.09** | 94.12 | 62.50 | 10.00 | 89.23 | 25.00 | 0.00 |
| + SST | **14.29** | 80.00 | 90.65 | 94.74 | 88.59 | 94.74 | 57.78 | **50.00** | 80.65 | **46.15** | 0.00 |
| + SST + SDS | 0.00 | 88.89 | 91.97 | 93.48 | 91.03 | **96.55** | 63.41 | 33.33 | 90.32 | 40.00 | 11.11 |

Table 10: Model comparison on various metrics (Part 2), including cash price (CshPrc), change price (ChgPrc), credit card price (CCP), e-money price (EMP), menu quantity count (MQtyC), menu type count (MTypC), total etcetera (TotEtc), total price (TotPrc), discount price (DscPrc), subtotal other (SubO), service price (SrvPrc), and subtotal price (STP).

performance and robustness, especially when using smaller guidance sets, the overall performance still falls short compared to mono-grained baselines. However, the proposed domain adaptation approaches significantly improve robustness when the guidance set size, $\mathbb{D}_n$, is reduced. In particular, Structural Domain Shifting (SDS) demonstrates a strong ability to capture domain-specific information across all guidance set ratios. Moreover, combining Synthetic Sequence Tagging (SST) with SDS results in even better performance when a larger, well-annotated guidance set is available.

| Model | 0% | 10% | 20% | 30% | 40% | 50% | 60% | 70% | 80% | 90% | 100% |
|---|---|---|---|---|---|---|---|---|---|---|---|
| Baseline | 0.00 | 59.73 | 67.45 | 78.36 | 80.54 | 76.34 | 78.86 | 78.86 | 78.02 | 79.70 | 81.21 |
| Joint-grained | 0.00 | 47.65 | 76.68 | 79.87 | 83.89 | 85.74 | 86.91 | 87.42 | 86.24 | 88.93 | 89.60 |
| + L2V | 0.00 | 48.83 | 76.68 | 85.57 | 84.23 | 88.93 | 87.42 | 86.58 | 87.92 | 89.93 | 90.60 |
| + SDS | **87.42** | **89.43** | **88.93** | 90.77 | 88.59 | 90.77 | 87.42 | 90.77 | **91.61** | 91.28 | 91.11 |
| + SST | 0.17 | 54.03 | 73.66 | 86.74 | 85.40 | 86.74 | 86.41 | 89.26 | 85.57 | **91.61** | 90.77 |
| + SST + SDS | 47.65 | 85.57 | 88.26 | **88.93** | 88.26 | 89.09 | 88.26 | **91.78** | 90.27 | 90.77 | **92.62** |

Table 11: Performance comparison of models at different guidance set ratios on printed set P.

### D.2.2 CORD DATASET

For the CORD dataset, different from the coarse-grained level task, integrating coarse-grained information into the fine-grained framework brings limited improvement.

### D.3 EFFECTS OF SYNTHETIC SET SIZE

In practical applications, the availability of synthetic document collections often depends on domain-specific factors. To evaluate the impact of varying $\mathbb{D}_n$ sizes, we analyzed how performance changes with different synthetic set sizes, as shown in Table 14 to demonstrate the effectiveness of the proposed framework. Generally, increasing $\mathbb{D}_n$ improves model performance during fine-tuning on $\mathbb{D}_g$. Domain adaptation methods that address structural domain shifts are less

| Config. | Form NLU | | Config. | CORD |
|---|---|---|---|---|
| | P | H | | |
| No DW | 89.60 | 85.76 | No DW | 88.11 |
| ½ SDS | 90.60 | 86.93 | ½ SDS | 89.27 |
| ½ SIT | 91.28 | 85.76 | ½ SST | 87.93 |
| ½ SDS+SIT | 90.60 | 85.59 | ½ SDS+SST | 88.25 |
| SDS | 91.11 | **88.78** | SDS | 89.08 |
| SIT | 90.77 | 87.94 | SST | 88.83 |
| SDS+SIT | **92.62** | 88.61 | SDS+SST | **90.25** |

Table 14: Effects of changing the size of synthetic annotated set $\mathbb{D}_n$

| Model | 0% | 10% | 20% | 30% | 40% | 50% | 60% | 70% | 80% | 90% | 100% |
|---|---|---|---|---|---|---|---|---|---|---|---|
| Baseline | 0.00 | 48.58 | 63.32 | 68.17 | 66.00 | 66.67 | 67.50 | 62.98 | 65.33 | 64.82 | 64.66 |
| Joint-grained | 0.00 | 36.85 | 71.86 | 73.37 | 71.19 | 82.91 | 84.25 | 82.75 | 82.41 | 85.59 | 85.76 |
| + L2V | 0.00 | 40.03 | 72.53 | 82.08 | 81.07 | 80.40 | 84.09 | 82.41 | 82.58 | 86.41 | 87.60 |
| + SDS | **81.74** | **85.26** | 86.93 | 82.91 | 83.39 | 85.26 | 84.09 | **89.45** | **87.94** | 87.77 | **88.78** |
| + SST | 5.70 | 44.39 | 67.17 | 81.24 | 76.55 | 83.25 | 84.09 | 87.94 | 84.09 | 87.10 | 87.94 |
| + SST + SDS | 44.22 | 82.41 | **84.59** | **87.44** | **83.56** | **85.26** | **84.76** | 86.77 | 85.09 | **89.45** | 88.61 |

Table 12: Performance comparison of models at different guidance set ratios on printed set H.

| Model | 0% | 10% | 20% | 30% | 40% | 50% | 60% | 70% | 80% | 90% | 100% |
|---|---|---|---|---|---|---|---|---|---|---|---|
| Baseline | 0.00 | 69.21 | **77.91** | 79.26 | 78.48 | 83.59 | 84.31 | 86.13 | 85.28 | 87.36 | 87.08 |
| Joint-grained | 0.00 | 68.57 | 75.77 | 78.68 | 79.24 | 83.33 | 84.03 | 86.24 | 85.01 | 86.98 | 87.48 |
| + L2V | 0.00 | 71.01 | 76.68 | 78.82 | 78.68 | 82.25 | 84.47 | 84.93 | 85.24 | 87.08 | 88.11 |
| + SDS | 0.05 | **72.03** | 77.85 | 77.10 | 78.69 | 84.83 | 85.21 | 86.41 | 85.84 | 88.20 | 88.81 |
| + SST | 0.25 | 63.73 | 71.21 | 73.32 | 76.31 | 81.26 | 82.37 | 83.03 | 84.91 | 87.76 | 88.78 |
| + SST + SDS | **4.21** | 68.61 | 77.67 | **79.34** | **80.31** | **85.35** | **85.22** | **87.38** | **86.48** | **88.25** | **89.33** |

Table 13: Performance comparison of models at different guidance set ratios on CORD dataset.

sensitive to $\mathbb{D}_n$ size, while methods like synthetic inquiry tuning and sequence tagging are more affected. This indicates that even a limited amount of synthetic structural information can effectively bridge domain gaps, though a larger $\mathbb{D}_n$ size further strengthens model robustness and overall performance.

### D.4 DAViD ROBUSTNESS ANALYSIS

To assess the robustness of the proposed framework and domain adaptation strategies, a synthetic label is introduced into the guidance set $\mathbb{D}_g$ of the CORD dataset. Instances are randomly selected based on a normal distribution, $X \sim \mathcal{N}(0, 1)$, with their corresponding ground truth label $y$ replaced by the randomly chosen label $\hat{y}$ from the label space $Y$ or assigned a "no" label ($\emptyset$). Adjusting the parameter $\lambda$, the synthesis ratio is controlled so that the proportion of noisy instances is given by $P(|X| > \lambda) = P_\lambda$. This allows for a thorough evaluation of the

| Model | $X \sim N(0,1), y \neq \hat{y}$ | | | $X \sim N(0,1), \hat{y} = \emptyset$ | | |
|---|---|---|---|---|---|---|
| | $P_2$ | $P_{1.5}$ | $P_1$ | $P_2$ | $P_{1.5}$ | $P_1$ |
| Baseline | 86.08 | 82.65 | 74.83 | 85.58 | 82.09 | 75.20 |
| Joint-grained | 85.47 | **82.81** | 74.45 | 86.21 | 82.79 | 76.40 |
| +SDS | 84.28 | 81.79 | 74.62 | 85.78 | 80.19 | **76.82** |
| +SST | 85.70 | 81.96 | 75.73 | 84.36 | 81.99 | 75.80 |
| +SDS+SST | **87.20** | 82.26 | **76.23** | **86.32** | **82.89** | 75.52 |

Table 15: Performance comparison of models under different types of synthetic annotation label (incorrect and incomplete) across varying synthesis ratios.

model's capacity to manage incorrect and incomplete labels across varying levels of label corruption.

**Robustness Analysis - Incorrect Labels** Incorrect label assignments cause models to learn inaccurate information during training, which could be used to assess their robustness in handling noisy or misleading data during training. As shown in Table 15, the joint-grained framework, warmed on SDS with NST, exhibits superior robustness compared to all other configurations, significantly outperforming the baseline. This highlights the effectiveness of the proposed frameworks and domain adaptation strategies in mitigating the negative impact of incorrect labels and enhancing model robustness in real-world applications.

**Robustness Analysis - Incomplete Labels** The absence or unavailability of labels prevents models from learning effectively from samples with missing labels, which is used as another criterion to assess the model's robustness in dealing with incomplete datasets. As shown in Table 15, joint-grained frameworks demonstrate consistent robustness compared to the mono-grained baseline model, highlighting that fusing coarse-grained information leads to a more robust fine-grained document representation. Additionally, after tuning the joint-grained framework on various domain adaptation tasks, the performance is further improved, illustrating that the proposed domain adaptation approach enhances robustness in scenarios where labels are absent.

## D.5 More Results and Analysis about LLMs/MLLMs testing.

### D.5.1 Prompt Details

The prompt details for each employed LLM/MLLM within the FormNLU dataset are provided in Table 16. The generated outputs are subsequently post-processed to compute the Jaccard distance between target entities, thereby ensuring accurate identification of the entity most closely matching the ground truth. For the CORD dataset, we adopt the LayoutLLM (Luo et al., 2024) configurations, utilizing ANLS as the evaluation metric.

| Model | Prompt | Image |
|---|---|---|
| GPT-3.5 | *Context: {} \n Above is the context of the target form document, please extract the {} \n, the output format strictly follow: Value: xxx* | N |
| GPT-4o-t | *Context: {} \n Above is the context of the target form document, please extract the {} \n , the output format strictly follow: Value: xxx* | N |
| LLAVA1.5 | *USER: Below image is the target form image. <image> \n Context: {} \n Above is the context of the target form document, please extract the {} only \n, the output format strictly follow: \n ASSISTANT:* | Y |
| QWen-VL | *Below image is the target form image. <image>\n Context: {} \n Above is the context of the target form document, please extract the {} only \n, the output format should strictly follow: \n Answer:* | Y |
| xGen-MM | *Context: {} \n Above is the context of the target form document, which is {} \n, output the answer only: \n Answer:* | Y |
| GPT-4o-v | *Below image is the target form image. <image> Context: {} \n Above is the document image and context of the target form document, please extract the {} \n, the output format strictly follow: Value: xxx* | Y |

Table 16: Comparison of prompts and image utilization across different LLMs/MLLMs.

### D.5.2 LLMs/MLLMs Performance Analysis

We show the breakdown performance of different LLMs/MLLMs predictions under zero-shot scenarios of printed set in Table 17 and handwritten set in Table 18, respectively. The results indicate that closed-source models exhibit relatively lower performance compared to other models. Consistent with the overall performance trends, closed-source models, even when utilizing non-multimodal output forms, tend to underperform against open-source MLLMs across most categories. Notably, the digit-based entities, e.g. ppn, pvp, located within the table remain challenging using text inputs alone, suggesting that incorporating visual information could enhance performance.

| Models | F1 | cnm | cid | hnm | hid | cdt | pdt | gdt | cls | ppn | pvp | cpn | cvp |
|---|---|---|---|---|---|---|---|---|---|---|---|---|---|
| GPT-3.5 | 34.37 | 96.00 | 88.00 | 47.92 | 17.00 | 32.00 | 30.00 | 66.00 | 96.00 | 0.00 | 4.00 | 12.00 | 4.00 |
| GPT-4o-t | 42.09 | **98.00** | **94.00** | 87.50 | 56.25 | 32.00 | 28.00 | 56.00 | **98.00** | 0.00 | 4.00 | 6.00 | 0.00 |
| LLaVA-1.5 | 9.79 | 10.00 | 72.00 | 10.42 | 16.67 | 0.00 | 8.00 | 20.00 | 12.00 | 0.00 | 0.00 | 46.00 | 0.00 |
| QWen-VL | 9.84 | 8.00 | 56.00 | 31.25 | 10.42 | 6.00 | 10.00 | 48.00 | 2.00 | 2.00 | 6.00 | 8.00 | 6.00 |
| xGen-MM | 12.62 | 46.00 | 6.00 | 12.50 | 22.02 | 26.00 | 10.00 | 40.00 | 34.00 | 4.00 | 14.00 | 34.00 | 6.00 |
| GPT-4o-v | 59.88 | 34.00 | 52.00 | 92.00 | 6.00 | 46.00 | 14.00 | **93.75** | 94.00 | **98.00** | 90.00 | 60.16 | 82.00 |
| Ours - Best | **92.62** | **98.00** | **94.00** | **95.83** | **79.17** | **86.00** | **80.00** | 92.00 | **98.00** | 92.00 | **96.00** | **98.00** | **98.00** |

Table 17: Zero-shot LLMs/MLLMs overall F1 and Breakdown Accuracy on FormNLU printed set. Explanation of abbreviations: cnm (Company Name/Scheme), cid (Company ID), hnm (Holder Name), hid (Holder ID), cdt (Change Date), pdt (Previous Notice Date), gdt (Given Date), cls (Class of Securities), ppn (Previous Person's Votes), pvp (Previous Voting Power), cpn (Current Person's Votes), cvp (Current Voting Power).

| Models | F1 | cnm | cid | hnm | hid | cdt | pdt | gdt | cls | ppn | pvp | cpn | cvp |
|---|---|---|---|---|---|---|---|---|---|---|---|---|---|
| GPT-3.5 | 30.94 | 86.00 | 62.77 | 58.00 | 18.74 | 20.00 | 16.33 | 34.35 | 90.00 | 4.94 | 10.12 | 31.00 | 6.17 |
| GPT-4o-t | 36.00 | 96.00 | 78.00 | 84.00 | 41.05 | 24.00 | 18.37 | 20.41 | **94.40** | 4.17 | 2.00 | 12.00 | 1.09 |
| LLAVA | 7.82 | 14.00 | 52.31 | 10.00 | 33.56 | 0.00 | 0.00 | 2.04 | 16.00 | 2.00 | 0.00 | 6.00 | 0.00 |
| QWen-VL | 6.00 | 8.43 | 36.00 | 20.00 | 24.00 | 20.00 | 6.12 | 18.37 | 2.00 | 2.00 | 4.08 | 2.00 | 8.00 |
| xGen-MM | 11.67 | 8.16 | 10.00 | 32.00 | 10.00 | 36.00 | 6.12 | 20.41 | 14.00 | 2.00 | 8.16 | 16.00 | 18.00 |
| GPT-4o-v | 49.15 | 98.00 | 29.59 | 54.73 | **97.14** | 39.78 | 24.15 | 26.00 | 78.77 | **96.00** | 20.18 | 48.06 | 5.41 |
| Ours - Best | **88.78** | **100** | **96.00** | **98.00** | 78.00 | **78.00** | **81.63** | **85.71** | 86.00 | 92.00 | **95.92** | **90.00** | **82.00** |

Table 18: Zero-shot LLMs/MLLMs overall F1 and Breakdown Accuracy on FormNLU handwritten set. Explanation of abbreviations: cnm (Company Name/Scheme), cid (Company ID), hnm (Holder Name), hid (Holder ID), cdt (Change Date), pdt (Previous Notice Date), gdt (Given Date), cls (Class of Securities), ppn (Previous Person's Votes), pvp (Previous Voting Power), cpn (Current Person's Votes), cvp (Current Voting Power).

### D.6 QUALITATIVE ANALYSIS: LIMITATIONS OF LLM/MLLMs

**Layout/Structure Interpretation** LLMs excel at processing unstructured text but struggle with understanding the spatial relationships and visual structures in form-based documents. This limitation results in misaligned content, missed logical groupings, and poor performance in tasks requiring precise layout comprehension, such as interpreting complex templates or extracting values from nested structures, as shown in Figure 8.

**Inconsistency** LLMs frequently produce inconsistent outputs when handling form-based documents, generating conflicting associations for the same key-value pairs or contradicting themselves across different sections. This lack of coherence highlights their difficulty maintaining logical consistency in structured content interpretation. For example, as shown in Figure 7, the LLM classifies differently between the exact same form or the same company forms with the same person's handwriting. The same limitation existed in the receipt dataset, CORD9.

**Lack of Contextual Understanding** LLMs often generate incorrect answers by relying on superficial patterns rather than understanding contextual relationships within the document. This results in confusion between unrelated elements, making LLMs unsuitable for accurately processing structured documents that require deeper contextual and spatial alignment, as shown in Figure 6

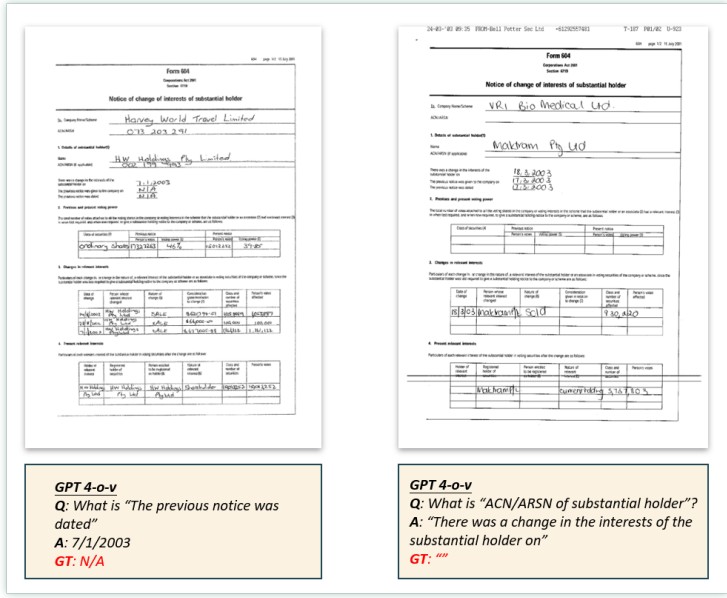

Figure 6: FormNLU sample with LLM-based document understanding (Lack of Contextual Understanding)

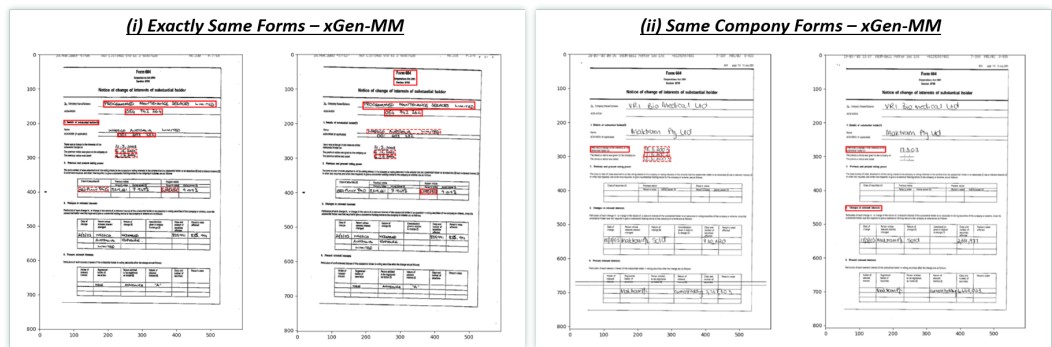

Figure 7: FormNLU sample with LLM-based document understanding (Inconsistency)

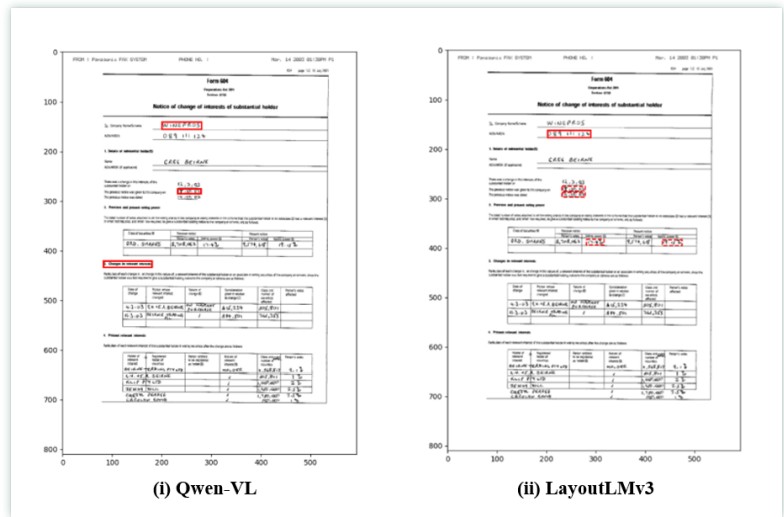

Figure 8: FormNLU sample with LLM-based document understanding (Lack of Layout Interpretation)

# E  SUPPLEMENTARY OF CASE STUDIES

Quantitative and qualitative case studies have demonstrated the effectiveness and robustness of the proposed joint-grained framework and domain adaptation methods. For further insights, additional supplementary materials and comprehensive analyzes are provided herein.

## E.1  SYNTHETIC LABEL SYNTHESIS DISTRIBUTION

As discussed in Section D.4, synthetic noise is introduced into the guidance set $\mathbb{D}_g$ of the CORD dataset. This noisy dataset is then used to fine-tune the model, which is subsequently tested on a well-annotated test set $\mathbb{D}_t$. Compared to the FormNLU dataset, the CORD dataset shows limited performance improvement. We applied random noise following a normal distribution to demonstrate the robustness of the proposed DAViD framework, rather than focusing solely on performance. This noise is introduced by replacing the original labels with incorrect labels (Figure 11) or marking them as unknown (Figure 12). Figures 11 and 12 illustrate the distribution of original and noisy labels across varying levels of noise rates.

## E.2  ADDITIONAL QUALITATIVE ANALYSIS

To highlight the strengths and weaknesses of the proposed DAViD framework, additional qualitative analyzes were conducted to compare the inference performance in a more straightforward manner.

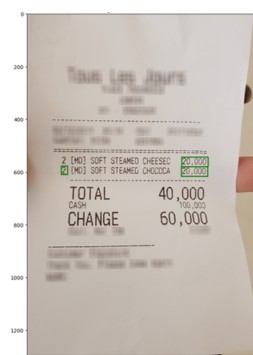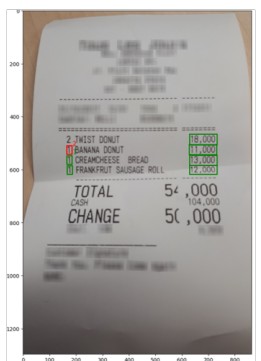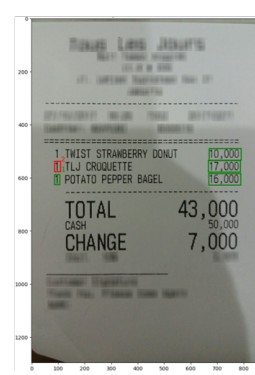

Figure 9: CORD LLM Case Study. (Inconsistency)

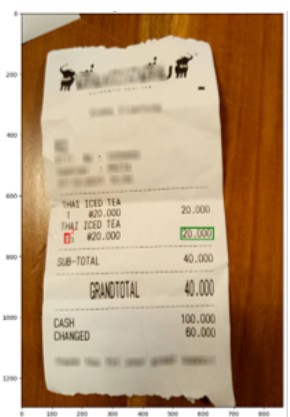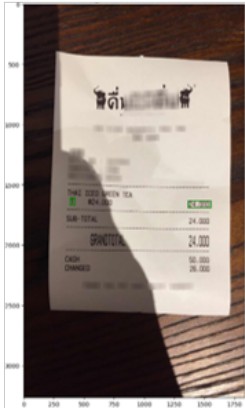

Figure 10: CORD LLM Case Study. (Inconsistency)

### E.2.1 QUALITATIVE ANALYSIS ON CORD

Additional visualized qualitative analysis samples are provided below, accompanied by more detailed descriptions in the captions.

### E.2.2 QUALITATIVE ANALYSIS ON FORMNLU

The visualized qualitative analysis for both the FormNLU printed and handwritten datasets is also presented. A more detailed analysis for each case is provided in the corresponding captions.

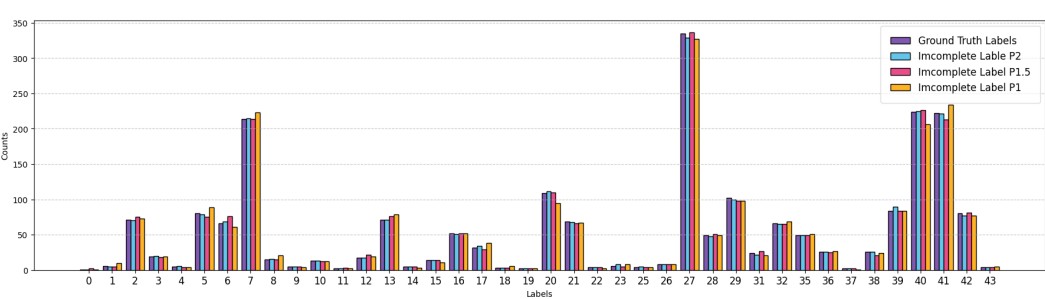

Figure 11: Comparison of category distributions in the ground truth and after applying varying levels of synthesis, where the ground truth labels are randomly replaced with another category following a normal distribution.

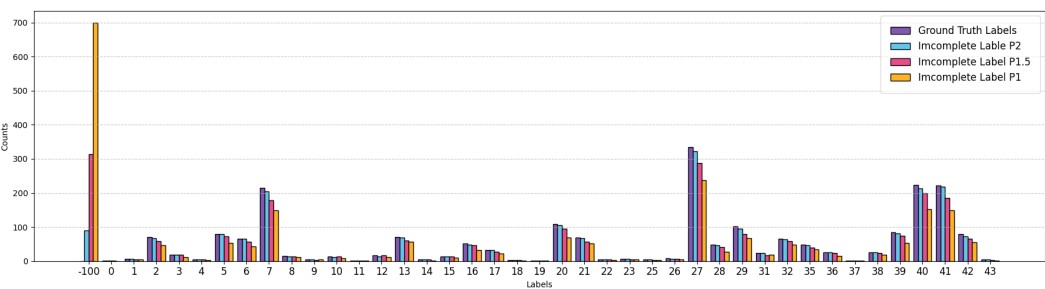

Figure 12: Comparison of category distributions in the ground truth and after applying varying levels of synthesis, where the ground truth labels are randomly replaced with unknown categories following a normal distribution.

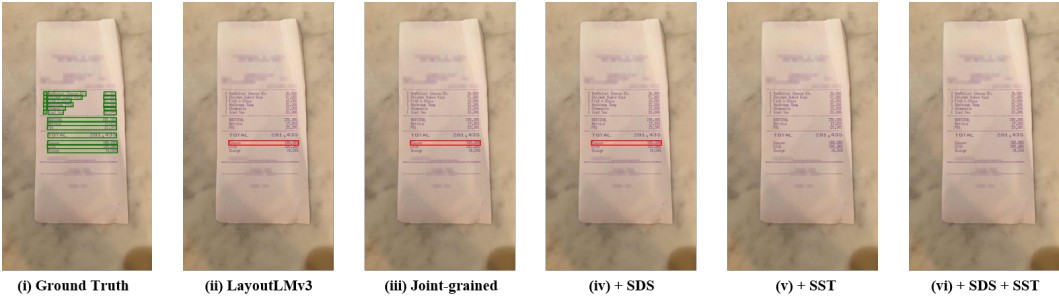

Figure 13: Real-world CORD dataset sample: (i) Ground truth key information highlighted in green. (ii) - (iv) Incorrect predictions marked with red rectangles under various configurations. (v,vi) The best performance was achieved **after applying SST** to extract all key information correctly.

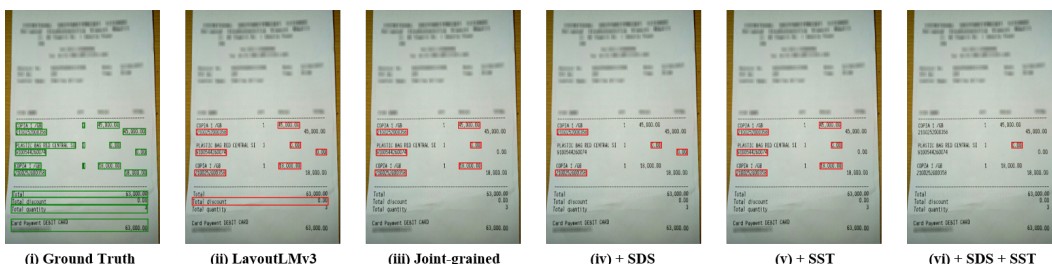

| (i) Ground Truth | (ii) LayoutLMv3 | (iii) Joint-grained | (iv) + SDS | (v) + SST | (vi) + SDS + SST |

Figure 14: Real-world CORD dataset sample: (i) Ground truth key information highlighted in green. (ii) - (vi) Incorrect predictions marked with red rectangles under various configurations. (vi) The best performance was achieved using two domain adaptation methods, with only one incorrect predictions. Compared to the fine-grained-only baseline LayoutLMv3, the Joint-grained framework effectively reduces the number of incorrect cases. The **application of SDS** further decreases erroneous predictions. While the number of errors remains unchanged after applying SST, **combining SST with SDS** improves robustness.

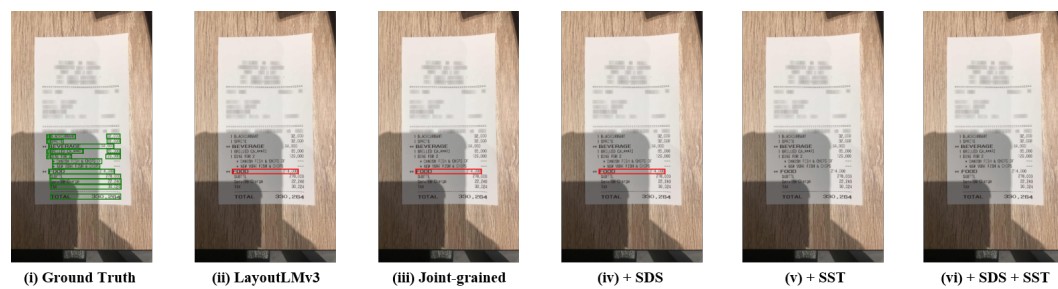

| (i) Ground Truth | (ii) LayoutLMv3 | (iii) Joint-grained | (iv) + SDS | (v) + SST | (vi) + SDS + SST |

Figure 15: Real-world CORD dataset sample: (i) Ground truth key information highlighted in green. (ii) - (iv) Incorrect predictions marked with red rectangles under various configurations. (v,vi) The best performance was achieved **after applying SST** to extract all key information correctly.

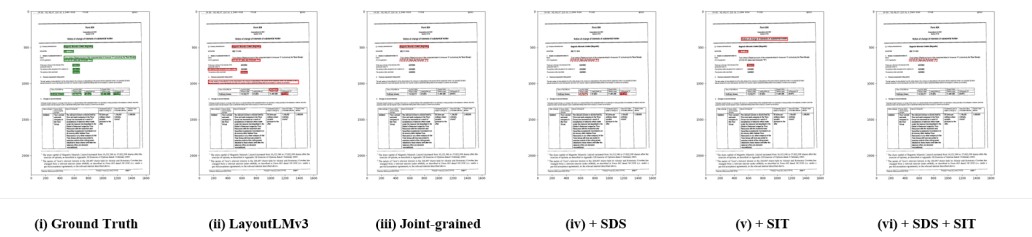

| (i) Ground Truth | (ii) LayoutLMv3 | (iii) Joint-grained | (iv) + SDS | (v) + SIT | (vi) + SDS + SIT |

Figure 16: Real-world FormNLU printed dataset sample: (i) Ground truth key information highlighted in green. (ii) - (vi) Incorrect predictions are marked with red rectangles under various configurations, and red dashed rectangles represent missing detection (unknown). The **joint-grained framework** significantly enhances performance on the target sample image by integrating fine-grained information into coarse-grained representations. While applying individual domain adaptation methods does not effectively reduce the number of error cases, combining both methods yields the best performance, with only one target entity value missing.

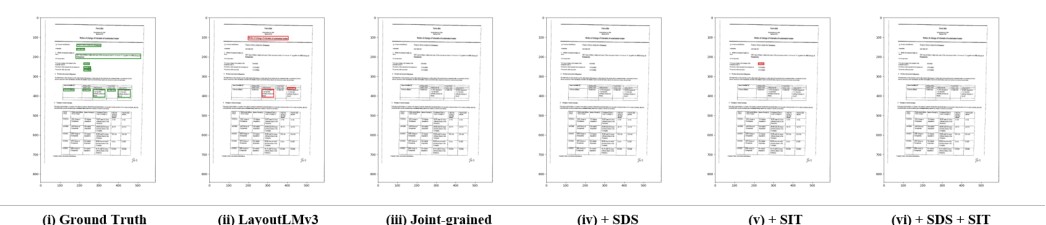

|  (i) Ground Truth | (ii) LayoutLMv3 | (iii) Joint-grained | (iv) + SDS | (v) + SIT | (vi) + SDS + SIT |

Figure 17: Real-world FormNLU printed dataset sample: (i) Ground truth target value entities are highlighted in green. (ii,v) Incorrect predictions marked with red rectangles under various configurations. Other configurations could detect all cases correctly, which may result from the effectiveness of **joint-grained** frameworks.

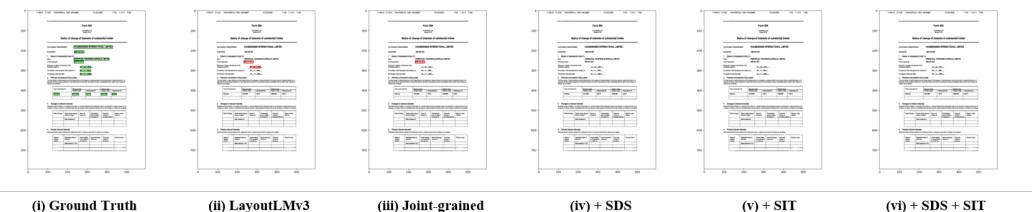

|  (i) Ground Truth | (ii) LayoutLMv3 | (iii) Joint-grained | (iv) + SDS | (v) + SIT | (vi) + SDS + SIT |

Figure 18: Real-world FormNLU printed dataset sample: (i) Ground truth key information highlighted in green. (ii,iii) Incorrect predictions marked with red rectangles under various configurations. The best performance was achieved using **any domain adaptation method**, resulting in no incorrect predictions.

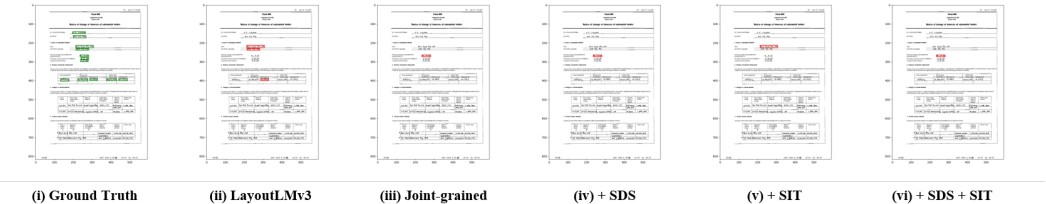

|  (i) Ground Truth | (ii) LayoutLMv3 | (iii) Joint-grained | (iv) + SDS | (v) + SIT | (vi) + SDS + SIT |

Figure 19: Real-world FormNLU handwritten dataset sample: (i) Ground truth key information highlighted in green. (ii) - (vi) Incorrect predictions marked with red rectangles under various configurations. **Joint-grained framework** could effectively reduce the number of incorrect predictions.

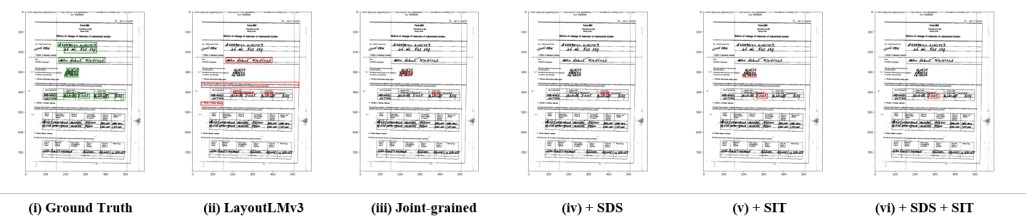

|  (i) Ground Truth | (ii) LayoutLMv3 | (iii) Joint-grained | (iv) + SDS | (v) + SIT | (vi) + SDS + SIT |

Figure 20: Real-world FormNLU handwritten dataset sample: (i) Ground truth key information highlighted in green. (ii) - (vi) Incorrect predictions marked with red rectangles under various configurations. A **joint-grained framework** significantly reduces incorrect predictions by integrating coarse and fine-grained features. The addition of **SDS** further enhances the prediction quality, resulting in more accurate and reliable outcomes.

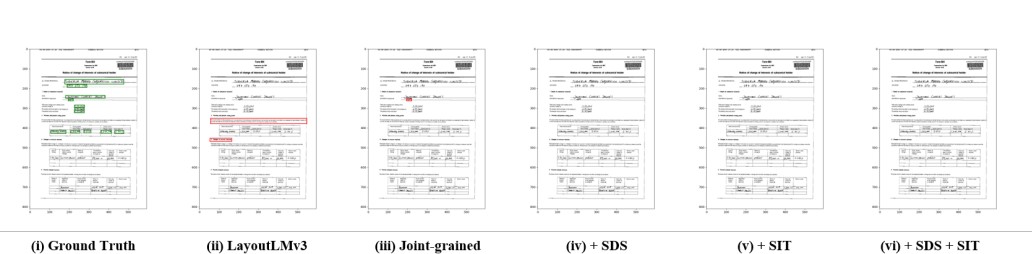

| (i) Ground Truth | (ii) LayoutLMv3 | (iii) Joint-grained | (iv) + SDS | (v) + SIT | (vi) + SDS + SIT |

Figure 21: Real-world FormNLU handwritten dataset sample: (i) Ground truth key information highlighted in green. (ii,iii) Incorrect predictions marked with red rectangles under various configurations. The best performance was achieved using **any domain adaptation method**, resulting in no incorrect predictions.

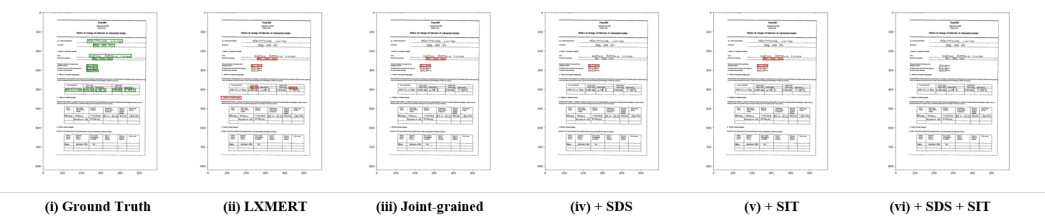

| (i) Ground Truth | (ii) LXMERT | (iii) Joint-grained | (iv) + SDS | (v) + SIT | (vi) + SDS + SIT |

Figure 22: Real-world FormNLU handwritten dataset sample: (i) Ground truth key information highlighted in green. (ii) - (v) Incorrect predictions marked with red rectangles under various configurations. (vi) The best performance was achieved using two domain adaptation methods, with no incorrect predictions. The **joint-grained framework** significantly enhances performance on the target sample image by integrating fine-grained information into coarse-grained representations. While applying individual domain adaptation methods does not effectively reduce the number of error cases, **combining both methods** yields the best performance without any incorrect prediction.

