# OpenReview forum: "DAViD: Domain Adaptive Visually-Rich Document Understanding with Synthetic Insights"
_ICLR.cc/2025/Conference — ICLR 2025 Conference Withdrawn Submission_

### Official Review · Reviewer_gvzr · 2024-10-20

**Soundness:** 3
**Presentation:** 2
**Contribution:** 3
**Rating:** 5
**Confidence:** 4

**Summary:**

The paper aims at solve the problem of labor consumption of well-annotated data for specific domain in visually-rich document understanding task. The paper proposed a joint-grained framework (token-level and entity level). To solve the lack of well-annotated data in specific domain, the paper proposes a method to utilize the LLM to tag the raw data. To adapt the method to unseen target domain as well as mitigate the gap between the LLM-labeled data and human-labeled data, the paper proposes a method to infuse the domain specific knowledge into model.

**Strengths:**

1.	The paper is well-motivated, and the proposed method can successfully infuse the domain-specific knowledge into existing method.
2.	The proposed method can be applied to different existing model, such as LayoutLMv3 and LXMERT, thus having generality to some extent.
3.	The experiment is well-organized, the paper conducts the extensive experiment to prove the effectiveness of proposed method and also compared with several large language model (LLM).

**Weaknesses:**

I have following concerns that need to be explained:
1.	When encountering a new domain, the proposed method requires collecting raw data, tagging a small portion manually, and using a large language model (LLM) to tag the majority. Then, the proposed pipeline is applied for training. This complex process may impede real application.
2.	The experiment does not test the method's performance in the original domain after training on new domain data, which is crucial to demonstrate the maintenance of knowledge from other domains.
3.	In Table 1, the experimental setup lacks clarity. It is unclear if "Full Training Set" refers to D_n and D_g. If so, the baseline method appears to outperform the proposed method, reducing the contribution of the method. Additionally, the evaluation metrics for Tables 1, 2, and 3 are not mentioned, causing confusion.
4.	In Section 6.2, the term "zero-shot" testing is misleading. The method uses some new domain data (a small portion labeled manually) to train the model, enhancing its ability in this domain. This is more suitable to be described as a few-shot application. Furthermore, the baseline settings in Table 3 are unexplained (likely LXMERT and LayoutLMv3) and the extreme low performance of the baseline method is also confusing. Domain knowledge is unlikely to significantly influence the performance, as the trained model should possess basic knowledge of the document understanding (DU) task. Clarification of the experimental setup is required to avoid confusion.
5.	The full name of L2V is absent from the paper. Given that an ablation study of L2V is conducted, indicating its importance, a detailed explanation is needed.
Additional readability issues are present:
1.	Many figures, especially Figures 1, 2, and 10, are blurred.
2.	Section 5.1 mentions handwritten (H), digital (D), and printed (P), but uses at least two abbreviations ((\mathcal{H}), (\mathcal{D}), (\mathcal{P})). Are these intended to have different meanings?
3.	In Table 2, it is unclear what FST stands for. Is it a typo or an unexplained technique?

**Questions:**

See above.

---

### Official Review · Reviewer_BgZL · 2024-10-30

**Soundness:** 3
**Presentation:** 2
**Contribution:** 3
**Rating:** 5
**Confidence:** 4

**Summary:**

This paper introduces a DAViD framework designed to achieve domain-adaptive rich visual document understanding (VRDU) by utilizing machine-generated synthetic data. The framework combines fine-grained and coarse-grained document representation learning and leverages synthetic annotations to reduce dependence on manual labeling. By utilizing pre-trained models and synthetic data, DAViD can achieve competitive performance even with minimal labeled datasets. Extensive experiments have validated the effectiveness of DAViD, demonstrating its efficient adaptability in specific domain VRDU tasks.

**Strengths:**

1. An innovative framework DAViD has been proposed for domain-adaptive document understanding, effectively utilizing synthetic data to reduce reliance on manual annotations.
2. By combining fine-grained and coarse-grained joint representation learning with large models, the performance and robustness of the model have been enhanced.

**Weaknesses:**

1.	The paper did not conduct experiments on advanced open-source large models, which limits a comprehensive assessment of the DAViD framework's performance across different types of models.
2.	The paper may not have discussed in detail the potential biases in the synthetic data generation process and how these biases could affect model performance.
3.	The paper focuses on solving the understanding problems of special domain documents, so the data selection should pay more attention to the diversity and professionalism of the field. The paper's discussion on the breadth of the data selected may not have been sufficiently explored.

**Questions:**

1. Regarding the comparison with LLaMa3: The paper mentions that the DAViD framework performs well on specific domain visual rich document understanding (VRDU) tasks, but it does not mention a comparison with existing advanced open-source large language models such as LLaMa3. Could the authors provide a performance comparison between DAViD and LLaMa3 on the same dataset? This would help readers understand the relative position of DAViD in the current research field.

2. Regarding the comparison with TextMonkey and DocOWL1.5: The paper does not mention comparative experiments with existing document-specialized multimodal large models such as TextMonkey and DocOWL1.5. Do the authors have plans or have already conducted comparisons with these models? Especially in the aspect of specific domain document understanding, these comparative results would be crucial for assessing the practical application potential of DAViD.

3. Regarding the generalization capability of the model: The DAViD framework focuses on efficient adaptability in specific domain VRDU tasks. Has the experiment considered the generalization capability across different types and complexities of documents?

---

### Official Review · Reviewer_7pqY · 2024-10-31

**Soundness:** 2
**Presentation:** 1
**Contribution:** 3
**Rating:** 3
**Confidence:** 3

**Summary:**

The paper presents the Domain Adaptive Visually-rich Document Understanding (DAViD) framework, designed to enhance information extraction from Visually-Rich Documents (VRDs), which typically include elements like charts, tables, and references. Traditional approaches require extensive annotated datasets, limiting their scalability due to labor-intensive manual labeling. DAViD addresses this by using machine-generated synthetic data for domain adaptation, along with fine-grained and coarse-grained document representation learning. This approach significantly reduces the dependency on manual annotations. Experiments demonstrate that DAViD effectively achieves competitive performance across domain-specific VRDU tasks with minimal annotated datasets, validating its potential as a scalable solution.

**Strengths:**

- The proposed approach seems effective and novel.

**Weaknesses:**

- A lot of unclear details: The most significant drawback of the paper is the poor writing that prevents the readers from appreciating the work. The paper introduces many components and creates many new terms. However many of them are unclear and not properly described. The new terms (e.g. L2V, SDS, SIT) could be better explained with figures (potentially Figure 1). But there was no illustration. See the questions below.

- Insufficient baseline comparison: The author should consider two additional baselines [1, 2]:


[1] Kim et al., 2022, OCR-free Document Understanding Transformer.

[2] Lee et al., 2022, Pix2Struct: Screenshot Parsing as Pretraining for Visual Language Understanding

**Questions:**

1. Line 137: what is KIE?
2. Section 4.1: why do we need two representations? can't we have the same representations and output of two granularity? if you need two representations, why do you need joint granularity extraction?
3. Line 188: what are standard tools?
4. How is GDE implemented?
5. Line 192: Simply referring the reader to Luo et al., 2022 is insufficient. It’s unrealistic to ask readers to read over a separate paper. Can you elaborate how you followed their work what you did?
6. How does L2V work, formally?

---

### Official Review · Reviewer_mAus · 2024-11-03

**Soundness:** 3
**Presentation:** 2
**Contribution:** 2
**Rating:** 3
**Confidence:** 4

**Summary:**

This paper proposes a Domain Adaptive Visually-rich Document Understanding (DAViD) framework, which utilizes the synthetic data to train some parts of the model for domain adaptation, and enhances the model’s performance on low-resource document understanding tasks. Extensive experiments are made to validate the effectiveness of the proposed method.

**Strengths:**

1.	The author proposes a joint-grained VRDU framework, which integrates fine-grained and coarse-grained document representations, leveraging pretrained models and synthetic data.
2.	The author proposes a synthetic data generation workflow that generates structural and semantic annotations using off-the-shelf tools and LLMs.

**Weaknesses:**

1.	The presentation of this paper is poor. The author introduce a lot of terms and abbreviations in Section 3 and 4. The method description is also redundant and complicated, which greatly hinders the readers from understanding this paper. Besides, I strongly suggest the author to rewrite the Introduction section, which is too redundant and repetitive. And I also recommend the author to replot Figure 1 and 3 into vector graphics, because the layout and small words are difficult to read.
2.	The author should list the number of parameters for all the method to compare different method’s computation resources. The author takes the document understanding (in this case, it’s key information extraction) into several parts, and uses synthetic data to pre-train each part. In this case where the number of parameters and the amount of training data are higher than the baseline, it is obvious that the final performance is better, especially under low-resource scenarios, which I think is quite trivial.
3.	In recent years, there are some MLLMs that are specially trained for document understanding tasks[1-3] and show stronger capability compared with old document analysis pre-trained models (LayoutLMv3, LiLT) and general MLLMs (Llava, Qwen-VL). I strongly suggest the author to add experiments on these MLLMs.

[1] Liu, Yuliang, et al. "Textmonkey: An ocr-free large multimodal model for understanding document." arXiv preprint arXiv:2403.04473 (2024).
[2] Wei, Haoran, et al. "Vary: Scaling up the vision vocabulary for large vision-language model." European Conference on Computer Vision. Springer, Cham, 2025.
[3] Hu, Anwen, et al. "mplug-docowl 1.5: Unified structure learning for ocr-free document understanding." arXiv preprint arXiv:2403.12895 (2024).

**Questions:**

1.	I want to know the size of each test set in Table 2.
2.	As the datasets used in this paper is too small (only a few hundred samples), I wonder whether the proposed method works on larger datasets (more than one thousand samples), which can also be seen as low-resource scenarios.

---

### Note · Authors · 2024-11-25

I have read and agree with the venue's withdrawal policy on behalf of myself and my co-authors.